# The spatial and temporal structure of neural activity across the fly brain

Evan S. Schaffer [1,8] ✉, Neeli Mishra [1,8], Matthew R. Whiteway [1,2], Wenze Li[1,3], Michelle B. Vancura[1], Jason Freedman[1], Kripa B. Patel[1,3], Venkatakaushik Voleti[1,3], Liam Paninski[1,2], Elizabeth M. C. Hillman [1,3,4], L. F. Abbott[1,5] & Richard Axel [1,6,7]

What are the spatial and temporal scales of brainwide neuronal activity? We used swept, confocally-aligned planar excitation (SCAPE) microscopy to image all cells in a large volume of the brain of adult *Drosophila* with high spatio-temporal resolution while flies engaged in a variety of spontaneous behaviors. This revealed neural representations of behavior on multiple spatial and temporal scales. The activity of most neurons correlated (or anticorrelated) with running and flailing over timescales that ranged from seconds to a minute. Grooming elicited a weaker global response. Significant residual activity not directly correlated with behavior was high dimensional and reflected the activity of small clusters of spatially organized neurons that may correspond to genetically defined cell types. These clusters participate in the global dynamics, indicating that neural activity reflects a combination of local and broadly distributed components. This suggests that microcircuits with highly specified functions are provided with knowledge of the larger context in which they operate.

What are the spatial and temporal scales of activity in the brain? In nematodes, flies, zebrafish, and mice[1], the exogenous activation of defined clusters of neurons can drive behavioral sequences, providing a causal link between the activity of small groups of cells and specific behaviors. In *Drosophila melanogaster*, defined clusters of genetically identified neurons can elicit innate behaviors, including aggression[2,3], courtship[4-6], and egg laying[7]. The identification of these circuits has suggested a view of the fly brain as a collection of specialized microcircuits. On the other hand, several locomotor behaviors seem to be associated with extensive activity in the fly brain beyond those neurons that are directly involved in the behavior. For example, locomotive behavior in the fly is associated with activity not only in motor circuits[8-10] but also in primary sensory areas[11-15] and downstream sensory structures such as the mushroom body[16,17]. These are similar to observations in mouse primary sensory cortices[18-20]. Thus, locomotor behaviors are often associated with more extensive patterns of activity than are required to elicit the specific behavior.

Brainwide recording of neural activity in multiple organisms reveals global activity associated with behavior[21-35] as well as cognitive tasks[36-38]. Recent studies in *Drosophila* have demonstrated extensive activity throughout most neuropil in the fly brain during running[25,26,39,40]. Similarly, both calcium imaging and electrophysiological recordings in the mouse have revealed distributed activity correlated with behavior across the cortex[30-32,34]. However, studies that employ neuropil imaging in the fly and widefield imaging in the mouse do not distinguish whether behavior results in the activation of all neurons or the activation of more limited but distributed clusters of neurons.

[1]Mortimer B. Zuckerman Mind Brain Behavior Institute and Department of Neuroscience, Columbia University, New York, NY 10027, USA. [2]Department of Statistics and the Grossman Center for the Statistics of Mind, Columbia University, New York, NY 10027, USA. [3]Department of Biomedical Engineering, Columbia University, New York, NY 10027, USA. [4]Department of Radiology, Columbia University, New York, NY 10027, USA. [5]Department of Physiology and Cellular Biophysics, Columbia University, New York, NY 10032, USA. [6]Department of Biochemistry and Molecular Biophysics, Columbia University, New York, NY 10032, USA. [7]Howard Hughes Medical Institute, Columbia University, New York, NY 10027, USA. [8]These authors contributed equally: Evan S. Schaffer, Neeli Mishra. ✉e-mail: ess2129@columbia.edu

Why does neural activity extend well beyond those neurons responsible for the behavior? Distributed activity may provide circuits that control specific behaviors with information relevant to the locomotor state of the organism. In this manner, sensory representations may also reflect behavioral state. For example, in visual systems, locomotion enhances gain and elicits activity in area V1 of mice[18,20] and in the optic lobe of flies[12,14,15]. Distributed activity associated with behavior may reflect efference copies that enable the cancellation of self-generated sensory input[41]. For example, locomotor state on the fly combines with self-generated visual feedback to control posture[42]. If a majority of neurons are indeed active during behavior, this would imply that the neural ensembles that are capable of eliciting specific behaviors (e.g., mating, aggression, or egg laying), will also be active during unrelated behaviors. This further implies that the ability of clusters of neurons to elicit specific behaviors must be modulated by behavioral context.

The fly brain offers a unique opportunity to examine the relationship between broadly distributed activity and the activity of spatially localized genetically identified neurons. Analysis of neural activity at both a global and local scale requires that we observe the activity of neurons distributed throughout the brain at sufficient temporal resolution to reveal correlations between neurons. We used SCAPE microscopy[43,44] to record activity in a significant fraction of the neurons across a large and contiguous portion of the brain of behaving *Drosophila*. The principal patterns of neural activity ("flygenvectors") comprise multiple spatial and temporal scales. We observe that signals related to some but not all behaviors engage the majority of imaged

neurons, including genetically defined neurons that control specific behaviors. Moreover, although the activity of most neurons is correlated with current behavior, a significant fraction exhibit activity correlated with behavioral dynamics on longer timescales, perhaps reflecting the animal's arousal state. The neural activity not explained by behavior is complex and high-dimensional, comprised of a large number of patterns distinguishable from noise. Most of these activity patterns are sparse and spatially organized, suggesting that each dimension corresponds to the localized activity of specific cell types. These groups of cells exhibiting unique local dynamics also participate in the global behavioral state, affording the opportunity for local computations to be state-dependent. Thus, neural activity in the behaving fly reflects the coordination of broadly distributed and spatially localized dynamics, and neurons with highly specified functions are provided with information about the larger behavioral context.

## Results

### Large-scale functional imaging at single-neuron resolution

We used SCAPE microscopy[43,44], a single-objective form of light-sheet microscopy that permits high-speed volumetric imaging, to examine activity across a large volume of the central brain of behaving adult *Drosophila*. This enabled dual-color imaging of the dorsal third of the central brain in the behaving fly at more than 10 volumes per second with a voxel size of $1.0 \times 1.4 \times 2.4\ \mu m$ (See Methods for details), greatly surpassing the spatiotemporal resolution of common methods such as two-photon imaging (Fig. 1a–b, g). We imaged flies expressing the nuclear calcium reporter nls-GCaMP6s and the static nuclear dsRed

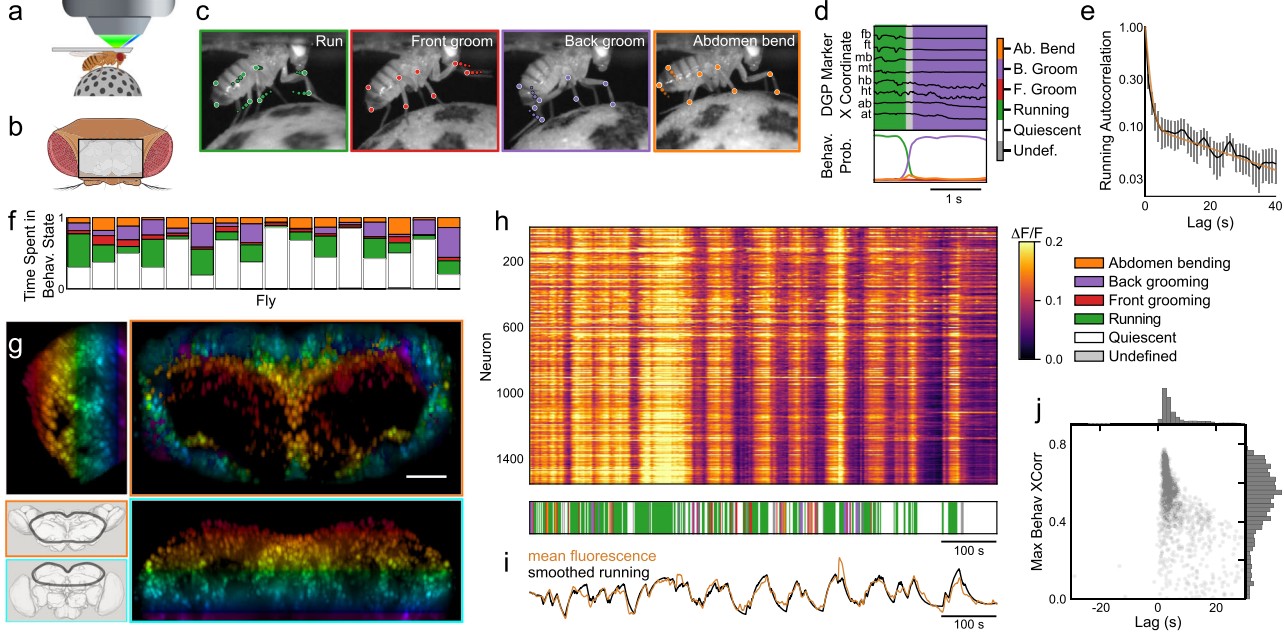

**Fig. 1 | Brainwide neural activity correlates with behavior. a** Illustration of SCAPE's imaging geometry. **b** The head of a fly viewed from a dorsal perspective (Top = posterior), with the approximate imaging window denoted by a black rectangle. **c** Points on the fly's limbs and body are tracked with Deep Graph Pose (DGP)[48]. Running, grooming, and abdomen bending exhibit distinct patterns of limb dynamics, observed in trajectories of DGP points. **d** A semi-supervised sequence model[49] extracts a timeseries of discrete behavioral states from DGP points. Example trajectories of the 8 tracked points shown in black above, ordered from anterior to posterior (fb: front bottom, ft: front top, mb: middle bottom, mt: middle top, hb: hind bottom, ht: hind top, ab: abdomen bottom, at: abdomen top). Inferred probability of each behavioral state is shown below, showing a transition from running to back grooming. The argmax of these state probabilities is shown in the ethogram above and hereafter. **e** The autocorrelation of running (black) is best-fit by the sum of two exponentials, with time constants of 1s and 40s (gold). Error

bars indicate ± SEM, $N = 18$. **f** Fraction of time spent in each behavioral state for each fly. Colors as in **d**. **g** Sample volume of raw imaging data in a brain with panneuronal expression of both nuclear-localized GCaMP6s and nuclear dsRed. Shown are maximum-intensity projections of the dsRed channel over the approximate dorsal/ventral (top right), anterior/posterior (bottom right), and medial/lateral dimensions (top left). Pseudocolor indicates depth in dorsal/ventral dimension. Scale bar in spatial map is 50 μm. Cartoon at bottom left shows the approximate location of the imaged volume on a reference brain from a dorsal (top) and anterior (bottom) perspective. **h** *Top*, raster of ratiometric fluorescence for all neurons from one fly (Fly 1 in **f**). Bottom, behavioral state, color coded as in **d**. **i** Average ratiometric fluorescence from all neurons (gold) and running smoothed with an exponential filter (black, time constant = 6s) are highly correlated ($r = 0.90$). **j** Maximum cross-correlation with running for every cell from the same fly as in **h**, versus the corresponding lag. Each point is one cell. **a**–**b** created with BioRender.com.

under control of the panneuronal driver nSyb-Gal4. Nuclear calcium reporters have been shown to be faithful readouts of neural activity[45,46]; they may preclude seeing fast dynamics and small changes in neural activity but offer the substantial benefit of easily resolving individual neurons. We therefore reasoned that any increase in low-pass filtering introduced by using a nuclear-localized indicator was greatly offset by the advantage of allowing cellular-level spatial resolution. We imaged a parallelepiped-shaped volume spanning the dorsal third of the central brain, achieving single-cell resolution through the majority of this imaged volume (Fig. S1). Kenyon cells were omitted from our analyses because nls-GCaMP6s expression was poor (Fig. S1). On the basis of cell counts from electron microscopy[47], we expected to resolve on the order of a few thousand cells (See Supplemental Information). We used the fluorescence of the static red channel to extract on average $1631 \pm 109$ ROIs per animal. After refinement to exclude ROIs with large motion artifacts, we obtained $1419 \pm 78$ stable, single-cell ROIs per animal (Methods). By visual inspection, we confirmed that this count contained nearly all neurons within $70 \, \mu m$ of the dorsal surface and a sample of neurons residing at greater depths (Fig. S1).

## Broad-scale neural activity is highly correlated with behavior

We examined neural activity while flies behaved freely on a spherical treadmill (Methods). The different behaviors exhibited by the fly were identified by tracking points on the fly's body with Deep Graph Pose[48]. We used a semi-supervised approach described in a companion manuscript[49] to infer the behavioral states of running, front and back grooming, abdomen bending, and quiescence (Fig. 1c–d, Methods). The average time flies spent in each behavioral state varied considerably (Quiescent: 50%, running: 19%, front grooming: 6%, back grooming: 15%, abdomen bending: 10%, undefined: 0.2%), and different flies exhibited these behaviors with varying frequencies (Fig. 1f). We also imaged the fly without a spherical treadmill, where it primarily exhibited a flailing behavior. When off the ball, flies flailed 12% of the time. On the treadmill, flies performed bouts of running punctuated by either grooming or quiescence. Autocorrelation of the running state decayed on time scales of 1s and 40s (Fig. 1e), because running occurred in bouts that lasted a few seconds but the tendency to run persisted for considerably longer times. The other annotated behaviors exhibited only a single fast correlation time (Fig. S1). Long-timescale changes in the tendency to run suggest that an underlying state, such as arousal, fluctuated over the course of our experiments.

Strikingly, most of the imaged neurons throughout the brain show a pattern of activity that is correlated with running. This is in accord with previous studies demonstrating that most of the neuropil in the fly brain is active when the fly runs[25,26,39,40] and demonstrates that these earlier neuropil recordings are not the consequence of a sparse ensemble of active neurons with extensive projections. Rather, running is represented by the vast majority of neurons in the fly brain (Fig. 1h). The mean activity across all the imaged neurons is highly correlated with running smoothed with an exponential filter with a decay time of 6s ($r = 0.90$, Fig. 1i). This correlation cannot be accounted for by motion artifacts; motion artifacts are negligible after registration, and movement of the brain before registration is not correlated with running ($r = 0.02$, Fig. S1). Cross-correlation of individual neurons with running is high, and the activity of most neurons follows running with a small lag (Fig. 1j).

## Distinct neural populations represent locomotion over different timescales

We fit a regression model to extract the components of neural activity correlated with all identifiable behaviors (running, front and back grooming, abdomen bending, and quiescence). Quiescence was characterized by a lack of movement of all tracked points on the body (Methods). Our tethered preparation prevented flies from exhibiting other behaviors such as proboscis or wing extension. To reflect moments of uncertainty in a fly's behavioral state, we used the behavioral state probability (Methods, Fig. 1d, bottom) rather than the binary behavioral state in our regression model. The observation that the autocorrelation of running exhibited two decay times (Fig. 1e) suggested that different neurons might be correlated with behavior on different timescales. Therefore, we regressed each neuron's activity against all behaviors filtered using a different fitted time constant ($\tau_i$) for each cell ($i$). We allowed for both potentially causal and acausal relationships between behavior and neural activity using a cell-specific temporal shift ($\phi_i$) of neural activity relative to the annotated behaviors (Methods). We assessed the significance of the fit to each cell by randomly shifting regressors in time (Methods).

Regressing neurons across behaviors and filtering each neuron with its own time constant considerably increased correlations between the activity of individual neurons and the annotated behaviors (Fig. 2a). This model accounted for proportionally more variance in flies that spent more time running ($CC = 0.73$, Fig. 2b), as expected from the widespread representation of running (Fig. 1h). The majority of neurons are positively correlated with running, although a smaller population show strong negative correlation with running (Fig. 2c). Negatively correlated neurons are highly concentrated in the Pars Intercerebralis (PI) (Fig. 2d–f). This region is comprised of a heterogeneous population of peptidergic neurons with a wide range of functions[50]. Although many PI neurons are anticorrelated with running, some PI neurons are positively correlated, suggesting that the release of a set of peptides is higher during running while release of others is higher during quiescence. Notably, many of these peptidergic PI neurons project to the same neuropil[50], meaning that this biologically meaningful heterogeneity in adjacent neurons would likely be masked in neuropil imaging.

Cells exhibited a remarkably broad range of preferred filter time constants (Fig. 2a, g). 41% of cells had small time constants ($\tau < 4$ seconds), reflecting the similarity of the dynamics of behavior and mean neural activity (Fig. 1e). However, 31% of all cells have $\tau$ greater than 20 seconds, and the overall distribution is bimodal (Fig. 2g). Thus, the neural relationship to behavior has two timescales that approximate the timescales of running itself (Figs. 1e and 2g). The median $r^2$ does not decrease as $\tau$ increases, indicating that behavior explains a similar fraction of neural activity in cells with small and large behavioral time constants (Fig. 2g). The temporal shifts in the filters were almost always positive and similar to the filter time constants, such that cells with large time constants also had large shifts (Fig. 2h). The locations in the brain of cells with a given behavior time constant exhibit spatial organization (Fig. 2i): some brain regions exhibit predominantly small $\tau$ and other regions exhibit large $\tau$. Neurons with large $\tau$ cluster in the PI region and in lateral areas on the posterior and anterior surfaces (Fig. 2i). Neurons with small $\tau$ are distributed throughout the brain but most concentrated near the midline on the dorso–posterior surface (Fig. 2i). This region is primarily composed of neurons innervating the protocerebral bridge and fan-shaped body of the central complex and descending neurons innervating the ventral nerve cord ("CX, DN", Fig. 2i, Fig. S2). This is consistent with the observation that neurons in these brain regions are involved in orienting and locomotion[51,52].

To explore whether neural activity might be related to aspects of behavior beyond those already considered, we refit the activity of every cell using the spatial coordinates of every tracked body point as a regressor, allowing for a unique behavior time constant ($\tau_i$) and temporal shift ($\phi_i$) for every cell as before (Methods). On average, the neural variance explained by this 'markers' model was higher than that of our original 'states' model, as expected given the significantly larger number of regressors (16 marker coordinates versus 4 active states, Fig. S2). However, the fraction of variance explained that exceeded expectation (from temporally shifted regressors. See Methods.) was similar for both models (Fig. S2), suggesting that our original model

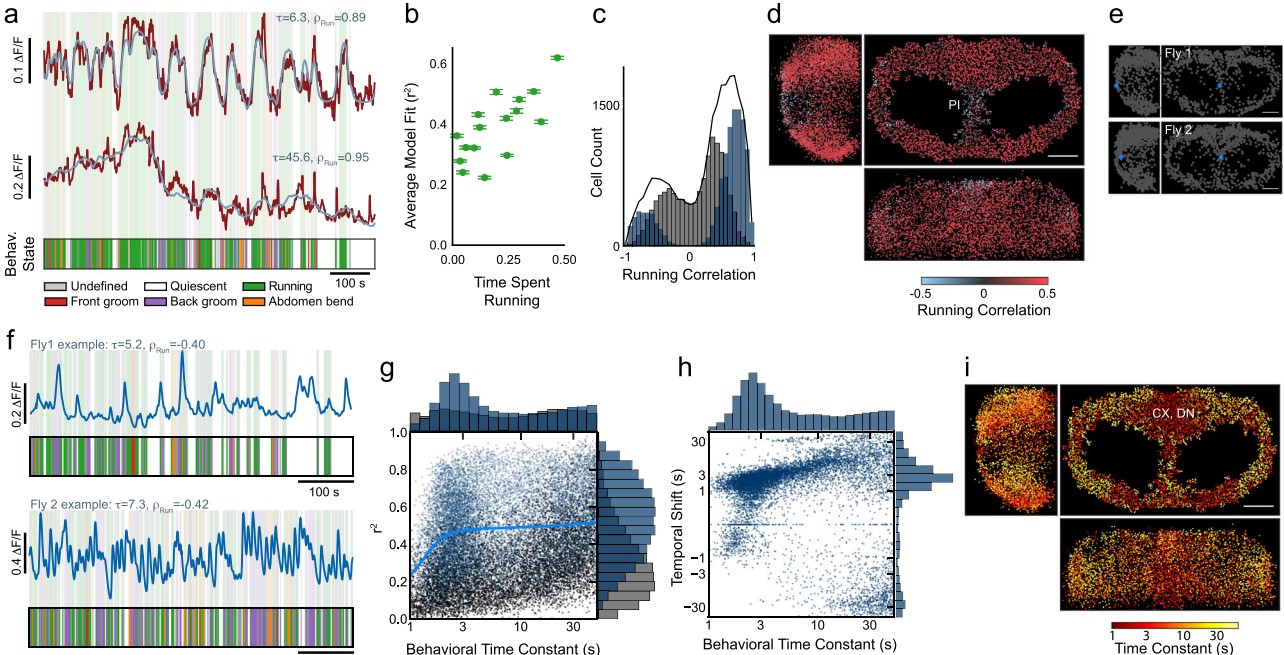

**Fig. 2 | Correlates of running are multimodal and spatially organized. a** Example traces from two cells, with regression fit overlaid in blue and ethogram below. **b** Average model fit across cells for each fly versus fraction of time spent running (Mean ± SEM, $N = 16$). Fly with highest time spent running shown in Fig. 1h–j. **c** Correlation with running for all cells and all flies ($N = 18$), cells significantly active during behavior in blue, all other cells in gray, total in black. **d** Downsampled composite spatial map of running correlation for all flies, viewed in the sagittal (left), transverse (right), and coronal (bottom) planes. **e** Location of an example cell (blue) in each of two flies (top and bottom, respectively) negatively correlated with running. **f** Corresponding activity traces for cells indicated in **e** for each fly. Etho-grams shown below for reference. **g** Distribution of behavior time constants ($\tau$) vs model $r^2$ for all flies ($N = 18$, cells significantly active during behavior in blue, all other cells in gray). Blue line indicates median $r^2$ of $\tau$ for all cells significantly active during behavior. **h** Distribution of $\tau$ vs distribution of time shifts ($\phi$) for all flies ($N = 18$), all cells significantly active during behavior. **i** Downsampled composite spatial map of $\tau$ for all flies, with large values in yellow and small values in red, viewed in the sagittal (left), transverse (right), and coronal (bottom) planes. Scale bar for all maps is 50 μm.

using behavioral states captures a relatively complete and more par-simonious relationship between neural activity and behavior. For these reasons, as we proceeded to examine relationships between neural activity and behavior, we exclusively focused on the more parsimo-nious 'states' model.

## Brainwide neural activity correlates with vigorous but not sub-dued behaviors

Do all behaviors engage the entire dorsal brain, or is running unique? Grooming and running are both precise directed behaviors but differ in the number of limbs they engage, whereas flailing and running both engage all limbs. We define behaviors engaging all limbs as 'vigorous' and behaviors engaging fewer limbs as 'subdued'. Most neurons are noticeably less active during grooming than running (Fig. 3a). During front and back grooming, only 3.0% and 2.1% of all cells, respectively, have $\tau < 4$s and a regression weight > 0.02 (Fig. 3b–c). Only 8 cells across all flies were highly correlated with front grooming ($CC > 0.5$, $\tau < 4$s), and only two flies had multiple such cells (Fig. 3d). In both flies, these cells were near the periphery of the imaged volume, potentially accounting for their absence in other flies. Flies engage in each beha-vior for different amounts of time, meaning that the variance explained by each behavior in neural data reflects both the duration and the influence of that behavior. Thus, to quantify brainwide influ-ence of each behavior, we normalize the variance explained by each behavior by the total time each fly exhibited that behavior, relative to running. Front and back grooming account for only 18% and 9% as much variance per unit time as running in the neural activity of cells with $\tau < 4$s (Fig. 3e, Fig. S3). Our observation that the dorsal brain is not broadly engaged during grooming is qualitatively in agreement with prior work proposing that small ensembles of cells are responsible for grooming[53,54].

We elicit flailing by removing the treadmill from beneath the fly. The representation of flailing is brainwide and qualitatively similar to that of running (Fig. 3f). 59% of neurons with regression weights > 0.02 and $\tau < 4$s during running had equally large regression weights during flailing (Fig. 3g, h). This suggests that global activity does not encode the precise modality of locomotion but rather may encode locomotive vigor or arousal more generally. This is further supported by the observation that unlike grooming, flailing accounts for more variance per unit time than running (218% and 262% for $\tau > 4$s and $\tau > 20$s, respectively. Figure 3e). Collectively, our results suggest that vigorous behaviors activate global representations, whereas more subdued behaviors such as grooming do not.

## Residual neural activity reveals ensembles of neurons with cor-related activity

We next examined the nature of the neural activity not accounted for by our regression model, and thus not easily explained by any of the identified behaviors. After large-scale locomotion- and other behavior-related activity has been regressed out, the residual activity exhibits rich dynamics across both space and time (Fig. 4a), with all neurons exhibiting significant residual dynamics across timescales, from sec-onds to minutes ($p < 1e{-}10$, Ljung-Box test. See Methods). This residual activity likely includes both activities unrelated to behavior as well as activity related to behavior but in a manner more complex than the regression model permits. For example, the residual activity of some cells appears to include dynamics related to transitions between states (Fig. 4a–b). We examined this by comparing neural activity preceding a state transition to activity earlier in a bout of a given behavior. On average, transitions from quiescence to running were preceded by a slight increase in residual neural activity (Fig. S4). However, we did not find evidence for a subpopulation of neurons that reliably encode state

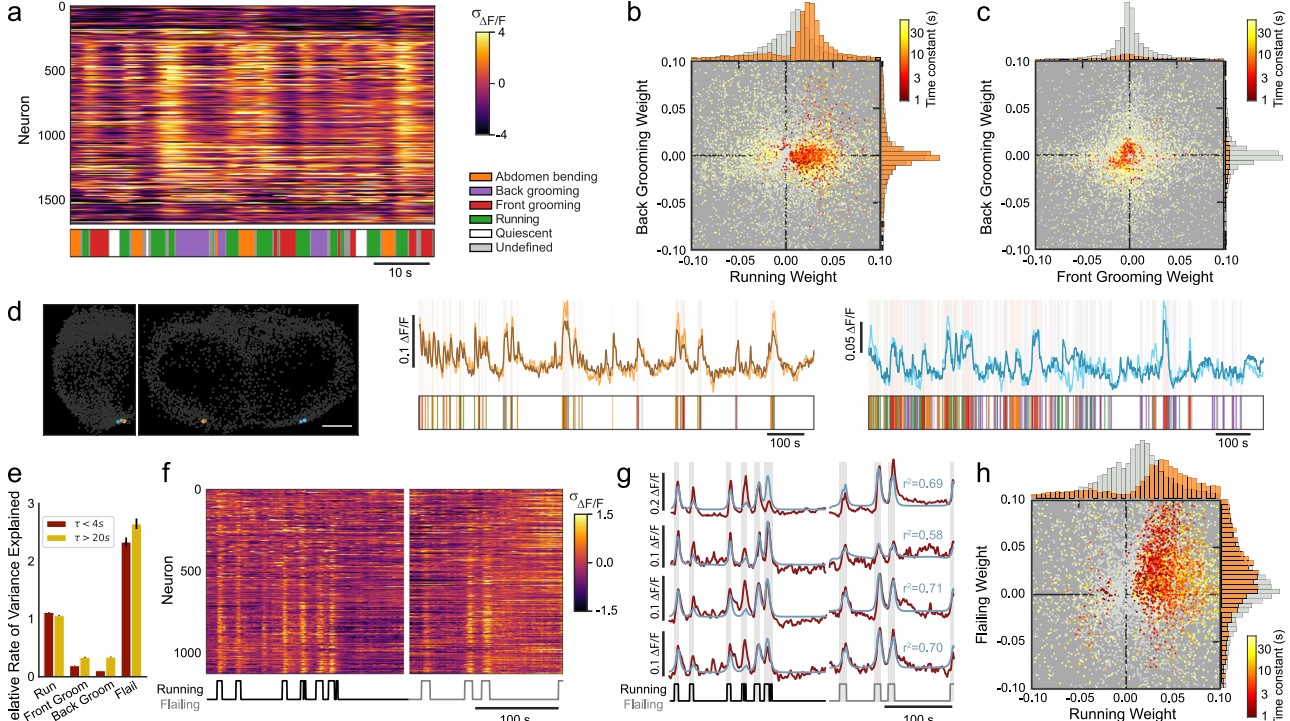

**Fig. 3 | Large-scale neural activity correlates with vigorous but not subdued behaviors. a** Example raster of z-scored $\Delta F/F$ for all cells from one fly in a short time window, showing individual bouts of many behaviors. Cells ordered by ascending $\phi$. **b** Regression weights for running vs. back grooming, for all flies ($N = 16$), all cells significantly active during at least one behavior colored by behavior time constant. Cells not significantly modulated during either behavior shown in gray. **c** Regression weights for front grooming vs. back grooming, for all flies ($N = 16$). **d** Left, location of pairs of cells in two flies (gold and cyan, respectively) correlated with front grooming. Scale bar is 50 μm. Right, corresponding activity traces for cells indicated at left for each fly. Ethograms shown below for reference. **e** Relative

rate of variance explained for each behavior, normalized to running, for all cells and all flies (For running and grooming: 17,404 large $\tau$ and 9812 small $\tau$ cells from 16 flies; For flailing: 8236 large $\tau$ and 4187 small $\tau$ cells from 10 flies). Error bars indicate ± SEM. **f** Raster of z-scored $\Delta F/F$ for all neurons from a fly running on a spherical treadmill (left) and then flailing in the absence of a spherical treadmill (right), with the timeseries of bouts of activity (running/flailing) shown below. **g** Activity from four example neurons (red) from the same fly as **f**, with regression model fits overlaid in blue and behavioral state (running/flailing/quiescent state) shown below. **h** Distribution of regression weights for running and flailing for all cells and all flies ($N = 10$).

transitions; any neuron is equally likely to exhibit a large response during the transition from one behavior bout to another (Fig. S4). For simplicity, hereafter we refer to activity accounted for and unaccounted for by the regression model as behavior-related and residual, respectively. On average, the fraction of variance explained by behavior (mean $r^2 = 0.39$) is similar in magnitude to that of the residual dynamics ($1 - r^2$). These residual dynamics include neurons that are highly active during running (Fig. 4b, red), and the variance explained by its leading PCs and behavior were negligibly correlated (Fig. S4). This implies that behavior-related and residual activity coexist in the same population of neurons.

We examined the structure of residual activity by performing a principal component analysis (PCA). On average, the first 10 modes explain 62% of the residual variance, and subsequent modes each account for no more than 2% of the variance (Fig. 4c). We quantified the dimensionality of this residual activity as the number of PCA modes that maximize the log-likelihood on held-out data. Higher-order PCA modes that do not improve the log-likelihood are not predictive of held-out data and therefore are defined as noise. Surprisingly, many modes can be distinguished from noise ($41.5 \pm 4.6$ modes, Fig. 4d, Methods), despite the fact that many of these modes account for very little variance. These PCA modes are very sparse, in some cases involving as few as 4 neurons (Fig. 4e, Methods). The average sparseness of the first two modes is 1.3%, meaning that a typical mode involves 18 neurons (Fig. 4f). Thus, modes that explain a small fraction of the total variance nevertheless describe reliable patterns present in neural activity. Counterintuitively, dominant modes are sparser than

less dominant modes (Fig. 4f). This suggests that the most reliable patterns in the data tend to contain fewer neurons.

Each PCA mode is sparse and therefore dominated by the activity of a small group of neurons with idiosyncratic yet similar dynamics (Fig. S4). These modes show spatial organization; for example, small groups of bilaterally symmetric neurons dominate the largest PCA modes (Fig. 4g–i). These modes are similar across flies, although there is variability in which mode explains the most variance in a given fly (Fig. 4g–i). To quantify this spatial organization, we first approximate each mode as a binary pattern in which only large outliers in the original mode are set to 1 (Methods). We then analyzed the spatial organization by calculating the distance between nonzero cells in the binary pattern after superimposing the left and right hemisphere by reflecting at the midline. Across all flies, modes were more spatially organized than expected by chance (Fig. 4j). The dominant modes identified by this analysis correspond to ensembles of ~20 cells that may comprise functional units. The ensembles often display symmetry across hemispheres. Each functional group is likely to be made up of multiple clusters with even smaller numbers of neurons, perhaps corresponding to specific cell types.

## Residual activity is similar in running and quiescent states

What is the relationship between global behavior-related activity and the sparser residual patterns of activity? One possibility is that residual dynamics could depend on behavioral state so that, for example, a particular residual dynamic pattern only appears during running (Fig. 5a, model 1). Alternatively, residual dynamics could be present in

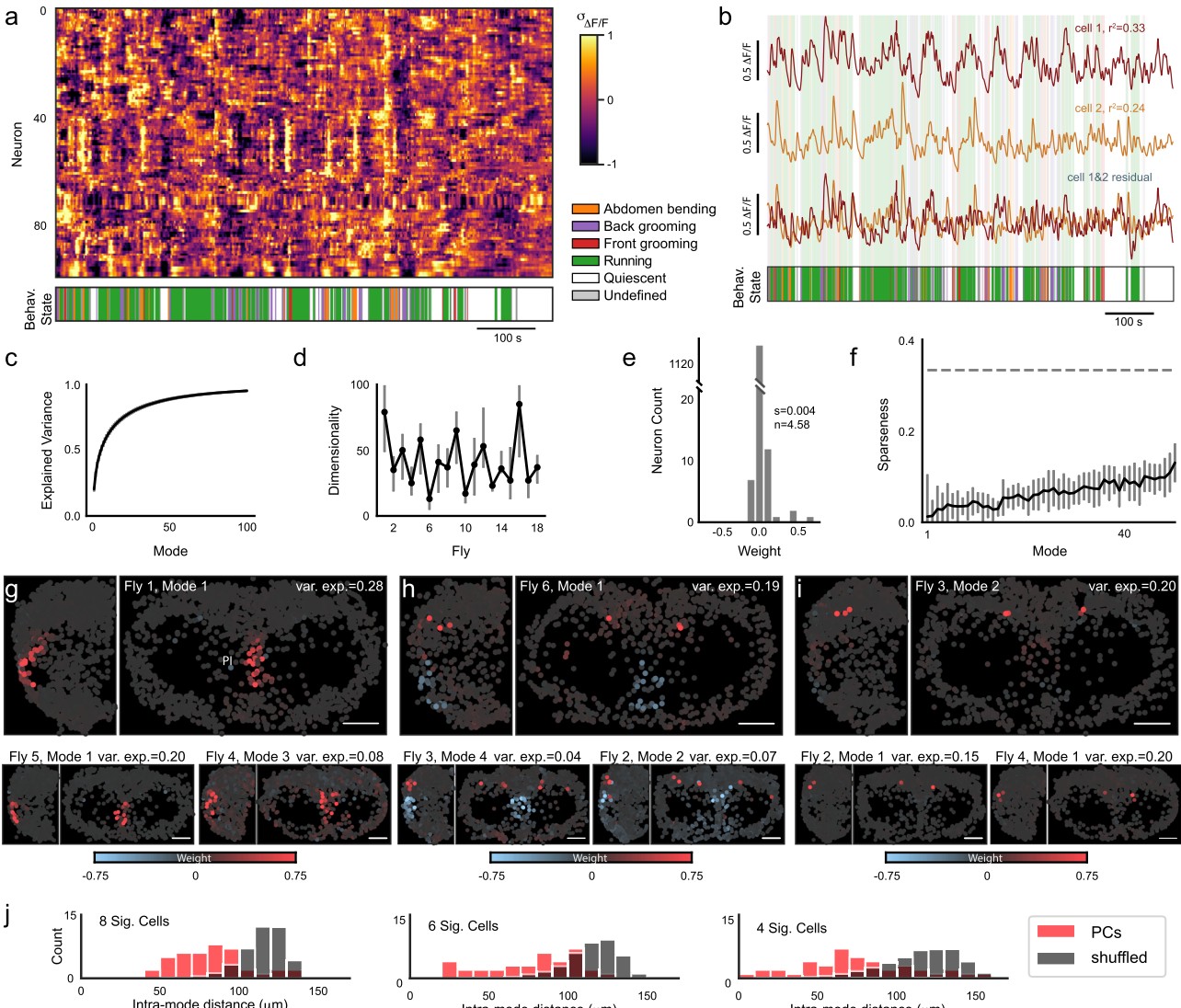

**Fig. 4 | Neural activity not accounted for by behavior is high-dimensional.**
**a** Example residual of the behavioral regression model reveals rich dynamics and groups of neurons with similar activity (z scored $\Delta F/F$, $N = 100$ cells, ordered by iteratively selecting the neuron most correlated with the previous neuron). Behavior ethogram shown below. **b** Example traces from two selected cells (red, gold, respectively) either before (top, middle) or after (bottom) subtracting the behavioral regression fit, with ethogram shown below. **c** The fraction of total variance explained in the regression residual as a function of the number of PCA modes (mean ± SEM, $N = 18$). **d** Dimensionality (number of modes) of the regression residual for all flies ($N = 18$), calculated as the peak in log-likelihood. Error bars indicate ± 1%. **e** Weights of all cells in a single representative PCA mode (fly 3, mode 2). Sparseness = 0.004, corresponding to 4.58 participating neurons. **f** Sparseness of

each PCA mode, averaged across all flies (Methods, median ± SEM, $N = 18$). Dashed line represents Gaussian zero-mean patterns. **g**–**i** Example maps of weights from leading PCA modes are sparse, approximately symmetric, and exhibit common patterns across flies (scale bar = 50 μm). Shown are examples dominated by Pars Intercerebralis (PI) neurons (**g**), dorso-posterior neurons (**i**), and anticorrelations between neurons from the two regions (**h**). Upper-right in **i** (Fly 3, Mode 2) is the same mode as shown in **e**. **j** Euclidean distance between cells with large magnitude PC components (red) for modes with 4, 6, or 8 such cells (left, middle, right, respectively) versus random groupings of cells of the same size (gray). Distance is computed after superimposing the left and right hemisphere by folding at the midline.

---

different forms in each of the multiple behavioral states (Fig. 5a, model 2). Finally, residual activity could be independent of behavioral state, and therefore similar, for example, in the running and the quiescent states (Fig. 5a, model 3). We find that the third of these possibilities most accurately accounts for our data; residual activity shows no obvious relationship to behavioral state (Fig. 4a).

We examined the residual neural activity during a behavioral state (a "subspace") and compared the subspaces of the running and quiescent states. The amount of variance explained by each mode appeared virtually identical in the two states (Fig. 5b). The dimensionality of these two subspaces is qualitatively similar, but on average the quiescent state is higher dimensional (37.9 ± 6.1) than the running

state (20.5 ± 2.0, Fig. 5c). This implies that the running and quiescent states are both complex.

We next asked if the residual activity during the running and quiescent states are not only similar in their complexity but also contain similar dynamics. We therefore determined whether the PCA modes defined in one state explain appreciable variance in the other state. PCA modes defined by activity during the quiescent state explain approximately 75% as much variance in the running state, and PCA modes of the running state explain 75% of the quiescent state (Fig. 5d). This implies that the subspaces occupied by the dynamics in each state are highly overlapping. Furthermore, the dimensionality of this overlap is similar to the dimensionality of

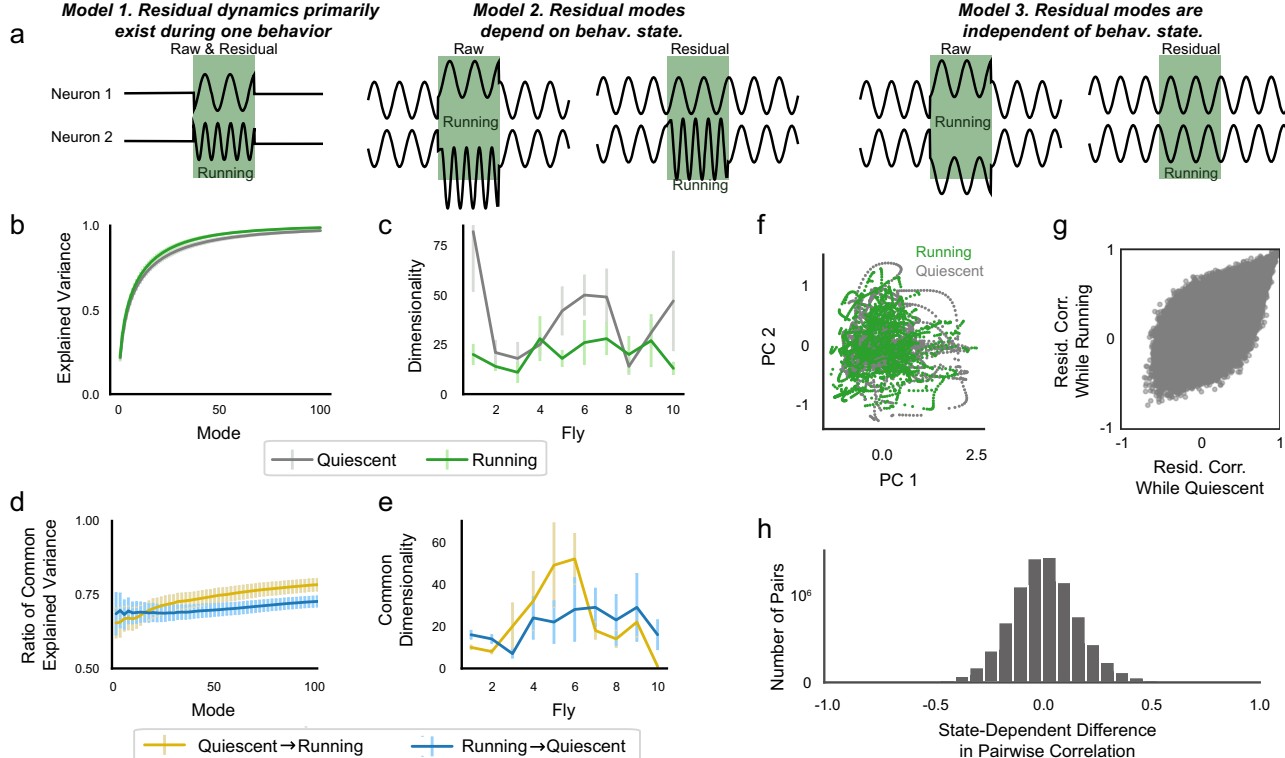

**Fig. 5 | Residual neural activity is largely independent of behavioral state.**
**a** Possible relationships between residual activity and behavioral state for two
cartoon neurons. Model 1: Residual dynamics only exist during one behavioral
state. Model 2: Both raw and residual dynamics depend on behavioral state. Model
3: Residual dynamics are independent of behavioral state. **b** Fraction of total var-
iance explained, as in Fig. 4c, but fitting and testing exclusively on times the fly was
quiescent or running (gray and green, respectively). **c** Estimated dimensionality for
the quiescent and running states, calculated as in Fig. 4d. **d** Using PCA modes
calculated as in **b** but evaluating them on the opposite behavioral state. Cumulative
variance explained in the opposite behavioral state is divided by variance explained
in the fitted behavioral state. **e** Shared dimensionality of the quiescent and running
states, calculated as in Fig. 4d. **f** Projection of residual dynamics during the running
(green) or quiescent (gray) states onto the first two PCs of the running state for an
example fly. **g** Residual pairwise correlation during either the quiescent or running
state, for all cells from one fly. **h** Distribution of differences of residual pairwise
correlations between the quiescent and running states for all flies ($N = 10$).

the activity (Quiescent-to-Running = 22.6 ± 5.1, Running-to-Quies-
cent = 20.8 ± 2.2, Fig. 5e). Moreover, projections of the residual
dynamics from both states onto the first two modes of the running
state are highly intermingled (Fig. 5f, also see Fig. S5). Collectively,
these results indicate that the temporal and spatial structure of the
residual activity is similar in the running and quiescent states.

PCA identifies patterns in the correlations across the full popula-
tion of neurons. To look for state-dependent effects in small groups of
cells, we compared correlations between the residual activity of all
pairs of cells in the quiescent and running states. These correlations
are similar with no large outliers (Fig. 5g, h). Thus, behavioral state and
the global pattern of activity associated with it appears to have only a
modest effect on the structure of residual activity. This is true not only
for the residual dynamics of large populations of neurons but also for
the residual correlations between all pairs of neurons (Fig. 5g, h). Thus,
behavioral state and residual dynamics appear remarkably indepen-
dent (Fig. 5a, model 3).

**Cluster analysis reveals spatially segregated groups of neurons
with correlated activity**
PCA revealed ensembles of spatially organized and functionally related
neurons in the residual activity. We identified smaller clusters of cor-
related neurons by performing hierarchical clustering analysis on the
residual activity (Fig. 6a). This procedure builds a tree of similarity
between the activity patterns of all cells, where at each branch point
the 'children' describe potentially meaningful subsets of a given 'par-
ent'. To look for structure in the data at all spatial scales without
defining arbitrary parameters for the number of expected clusters, we

identified significant clusters using cross-validation (Methods). Speci-
fically, we determined whether the variance of each child cluster was
significantly smaller than the variance of random samples of the same
size extracted from the parent cluster (Methods). In this way, we
determined whether a given small group of neurons defined a cluster
unique from other members of the parent cluster. Both a child and its
parent cluster can be significant, and therefore neurons may partici-
pate in dynamics organized on multiple spatial scales.

Figure 6a shows the full clustering tree for one fly, with each
branch colored according to whether the parent was a significant
cluster (not significant in black, all other colors significant). We
observed significant clusters of many sizes, including one cluster
comprised of more than half of all neurons but also many clusters
comprised of only two neurons (Fig. 6a–b). We next asked whether
significant clusters are spatially organized. A subset of Pars Inter-
cerebralis neurons located near the midline form a spatially compact
cluster that is identifiable across flies (Fig. 6c–d, white). Significant
clusters that share a parent with the Pars Intercerebralis cluster are
predominantly in posterolateral regions (Fig. 6c–d, yellow). Thus,
there is spatial organization and stereotypy at multiple spatial scales.
The full distribution of sizes for all significant clusters (Fig. 6e) reveals
a large number of significant clusters with 2 members. These clusters
exhibit diverse residual dynamics, but each cluster consists of pairs of
cells with similar dynamics (Fig. 6g). Despite these clusters being de-
fined by residual dynamics, neurons in the same cluster have a similar
relationship to global activity and behavior (Fig. S6). As a population,
cells within a cluster exhibit a distribution of behavioral time constants
and correlations indistinguishable from the distributions across all

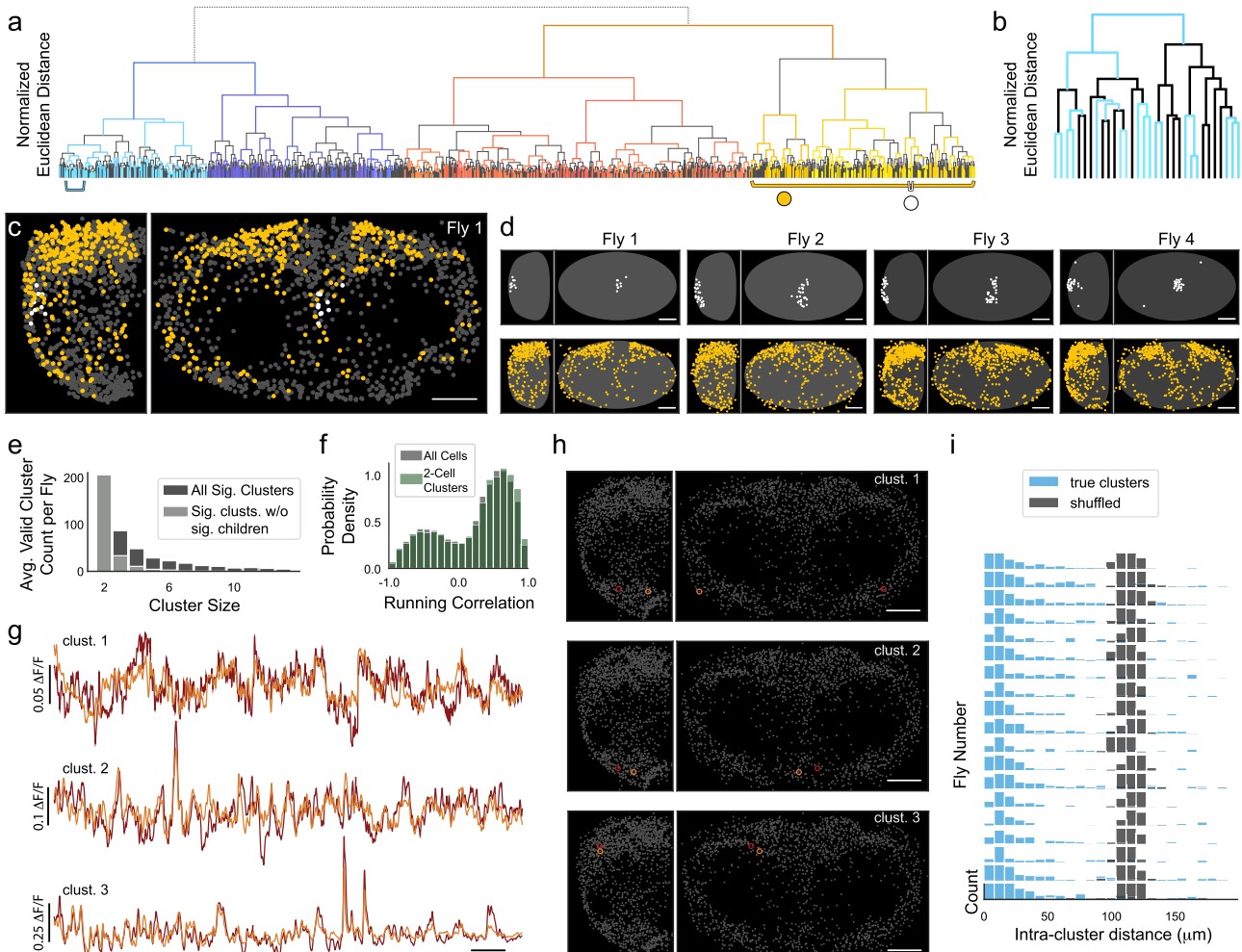

**Fig. 6 | Residual neural activity is composed of organized clusters on multiple spatial scales. a** The relationship between all cells in one fly, defined by hierarchical clustering on residual neural activity. Vertical axis reflects relative Euclidean distance in activity space, with the exception of the topmost dashed line, which is not to scale. Significance of each cluster was assessed by comparing the variance of the child cluster to the variance of samples from the parent cluster on held-out time points. Branches from non-significant clusters colored black, branches from significant clusters in other colors. Cyan bracket under the tree indicates region shown in **b**, and yellow and white markers under the tree indicate clusters highlighted in **c**. **b** Magnification of the portion of the tree indicated by a cyan bracket in **a**. **c** Example map of cluster identity for PI cluster (white) and neighboring clusters (yellow), with identity indicated by markers in **a**. **d** Same as **c** for additional flies, with fly 1 repeated for clarity. **e** Distribution of the size of all significant clusters (dark gray) and significant clusters that have no significant children (light gray). **f** Distribution of correlations with running for all cells (gray) and cells belonging to a significant two-cell cluster (green). **g** Residual neural activity from three example clusters each comprising two neurons. **h** Cells belonging to clusters shown in **g** in red and gold, with non-member cells in gray. Scale bar is 50 μm. **i** Euclidean distance between cells belonging to a 2-member cluster (blue), versus randomly assigned cluster labels (gray). Distance computed after superimposing the left and right hemisphere by folding at the midline. Each row shows a different fly and bar height is capped at 30.

cells (Fig. 6f, S6). Thus, clusters are highly diverse and participate in the global behavioral state.

By visual inspection, many small clusters appear to be either bilaterally symmetric or spatially localized (Fig. 6h). To quantify this observation, we analyzed the spatial organization by calculating the distance between cells in a cluster after superimposing the left and right hemisphere by folding at the midline (Methods, Fig. S6). Most clusters with two members were more spatially organized than expected by chance (Fig. 6i, S6). The presence of small clusters that are predictive of both activity patterns and spatial location is consistent with the association of cluster identity with function—cells with similar dynamics and similar function are likely to be in similar locations. These observations suggest that the fly brain is composed of many small subpopulations that collectively account for the high dimensionality of the brainwide data. Two-member clusters are embedded in larger ensembles of neurons, implying that the functional relationship between neurons is hierarchical. This is

consistent with known classes of cells in the fly brain—for example, Kenyon cells can be subdivided into $\alpha/\beta, \alpha'/\beta'$, and $\gamma$ subclasses; similarly, dopaminergic neurons can not only be divided into subclasses such as the PPL1 cluster but the PPL1 cluster can be further divided into single identified neurons that innervate distinct mushroom body compartments[55].

Our functional profiling of the brain offers a novel and complementary method of identifying cell types throughout the brain. The vast majority of cells in the central brain can be transcriptionally characterized as consisting of a few thousand distinct cell types that come in clusters of 1-10 neurons per hemibrain[56]. Histograms of the number of cells within each cell type from genetic and connectomic cell-typing[56] show an exponential shape similar to that revealed by our activity-based analysis (Fig. 6e). Thus, the smallest spatially organized subpopulations we identified functionally may correspond to genetically defined cell types.

## Egg-laying command neurons correlate with running

In the fly, small identified circuits that control specific behaviors have been elucidated. Our observation that most neurons in the fly brain are active during running and flailing suggests that neurons engaged in specific behaviors, such as mating, aggression, or egg-laying, are also active during spontaneous running. To test this, we asked whether the recently identified oviDN egg-laying command neurons[7] are active during locomotion. We imaged flies expressing the nuclear calcium reporter nls-GCaMP6s and the static nuclear dsRed under control of the split-GAL4 oviDN-SS1[7], which cleanly labels two of the three oviDN neurons in each hemisphere (Fig. 7a). We did not observe egg-laying behavior while flies were on the ball and thus, as expected, oviDN neurons exhibited little activity while flies were in the quiescent state. Neural activity was reliably higher during bouts of running (Fig. 7b), and running accounted for substantially more variance in the neural data than expected by chance ($p < 0.05$), consistent with previous work[57]. As observed in our panneuronal imaging data, total variance explained was highly correlated with time spent running (Fig. 7c), suggesting that heterogeneity in the neural data is accounted for by heterogeneity in the behavior. Thus, as predicted by our panneuronal data, neurons with highly specified function are provided with knowledge of the larger context in which they operate. This knowledge is reflected in the activation of egg-laying neurons, and therefore gating mechanisms are required to ensure that behaviors occur at the right time and place.

## Genetically defined subpopulations of PI neurons are inversely correlated with running

Clusters of cells with similar activity may correspond to genetically defined cell types in the fly brain. To explore this, we focused on cell types within the PI region. Panneuronal imaging revealed neurons in this region anticorrelated with running, in sharp distinction to the majority of imaged neurons (Fig. 2d, e). We examined the activity of two peptidergic cell types within PI, Dilp and Dh44, the latter a subset of the former[58]. Consistent with expectation from panneuronal imaging, many Dilp and Dh44 neurons showed an inverse relationship with running (Fig. 7d–f). Indeed, analysis of the distribution of running correlations observed in different parts of the brain confirmed that Dilp and Dh44 exhibit running correlations that one would only expect to find in PI (Fig. S7). Thus, these cell types are likely to correspond to unique clusters of neurons we identified in PI with panneuronal imaging.

## Discussion

We used SCAPE microscopy to record from a large volume of the dorsal brain with cellular resolution, complementing large-scale studies of neuropil regions in the fly brain[25–27,36,39,40,59]. To achieve cellular resolution, we used nuclear calcium as an indicator of neural activity. Trafficking of calcium into the nucleus is regulated by neural activity and influences gene expression[60]. SCAPE imaging permitted us to record from all neurons in a contiguous and large brain volume at high speed, providing an extensive picture of the neural correlates of behavior with cellular resolution. When placed on a ball, flies run, groom, or are quiescent. When suspended, flies often flail. Running and flailing engage a large fraction of the neurons in the imaged volume. A much smaller fraction of the neurons exhibit activity correlated with grooming. These behaviors unfolded over seconds and minutes (Fig. 1e), giving us the opportunity to resolve the neural correlates of these timescales. A regression model reveals neural activity correlated with running on both short and long-time scales. This suggests that most neurons are correlated with the act of running, and a significant fraction are correlated with the tendency to run. Moreover, cells with a given behavioral time constant are spatially organized, in some cases aligning with areas known to be involved in metabolism or locomotion. For example, a region we observed to have activity most highly correlated with behavior aligned with boundaries of specific cell types innervating the central complex (Fig. 2i, S2). More generally, the identity of neurons in each functionally defined region is unknown but can be loosely constrained by cell body locations in existing anatomy databases[47,61].

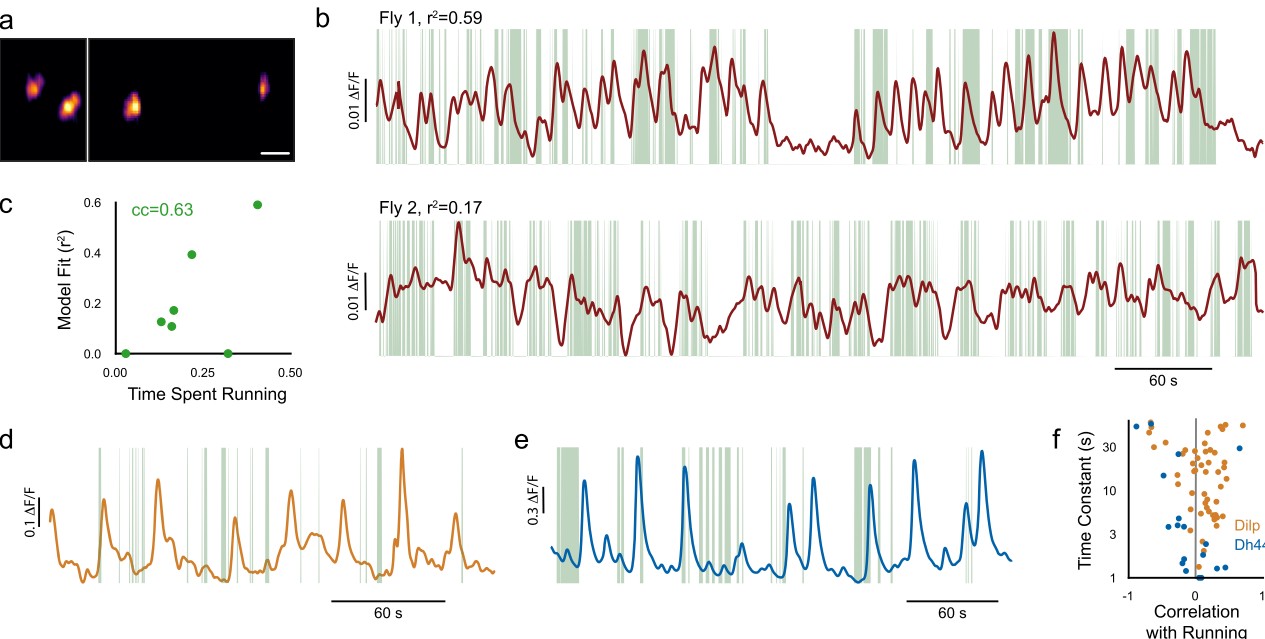

**Fig. 7 | Activity of defined cell types correlates with running. a** oviDN-SS1 (split-GAL4) labels a pair of neurons in each hemisphere. Scale bar is 50 μm. **b** Activity of oviDN neurons for two example flies, with bouts of running indicated in green. **c** Model fit for each fly versus fraction of time spent running ($N = 7$), as in Fig. 2b. **d** Activity of a Dilp neuron, with bouts of running indicated in green. **e** Activity of a Dh44 neuron, with bouts of running indicated in green. **f** Distribution of behavior time constants and correlations with running for all Dilp (57 neurons, 7 flies) and Dh44 (20 neurons, 5 flies) neurons whose behavior time constant was less than 60s (57 of 77 and 20 of 24, respectively).

Subtracting the dominant activity correlated with behavior reveals additional rich dynamics across time and space. This residual activity likely includes both activity unrelated to the exhibited behaviors as well as activity related to behavior but in a manner more complex than our regression model permits. Interestingly, this activity shows little dependence on locomotive state: residual activity exhibits similar spatiotemporal patterns in running and quiescent states. Thus, local computations appear to be superimposed upon a global behavioral state but not strongly state-dependent. This is similar to the observations that behavior-related activity is widespread but orthogonal to other dynamics in dorsal cortex of the mouse[31], and that preparatory and muscle-related activity are orthogonal to one another in the primary motor cortex of the monkey[62].

Neural activity not accounted for by behavior is high dimensional and sparse. Hierarchical clustering reveals small groups of neurons with highly correlated activity, at the extreme comprised of only 2 cells. These functionally defined clusters may correspond to genetically defined cell types in the fly brain. Consistent with this expectation, genetically defined cell types can account for the clusters we observed with panneuronal imaging in the Pars Intercerebralis (Fig. 7d–f). These small circuits do not operate in isolation. Clusters defined by the residual activity also participate in the global behavior-related dynamics. Thus, global patterns may inform local computation and in turn, local computations may influence global patterns.

The global scale of neural activity correlated with locomotion in flies is consistent with findings in worms[21,22,33], zebrafish[23,24,35] and mice[30–32,34]. Studies in flies[25–28,36,39,40] and those in other organisms pose the question of the mechanism and function of broadly distributed brainwide activity. In the fly, small identified circuits that control specific behaviors have been elucidated. However, we have shown that most neurons in the fly brain are active during running and flailing, either as actors or observers. This suggests that neurons engaged in specific behaviors, such as mating, aggression, or even egg laying, are also active during spontaneous running, without the act of running triggering these other behaviors. Indeed, we find that egg-laying command neurons[7] increase their activity during running without eliciting egg-laying. Downstream circuits must therefore be gated by behavioral state.

The brainwide behavioral state could arise from a variety of sources. For example, global activity could arise from widespread neuromodulation. Alternatively, the recurrent connectivity of the fly nervous system could provide a pathway for this global activity. One of the most plausible sources is the extensive afferent input to the brain from the ventral nerve cord—~2500 neurons originating in the ventral nerve cord project diffusely to the central brain[52]. Subsets of these neurons have recently been shown to encode behavioral states[63].

We observe a small but substantial fraction of neurons that correlate with locomotion on timescales longer than the duration of individual running bouts. These neurons may represent a locomotor state, the tendency to run. Many of these neurons reside in large posterolateral clusters and in the dorsomedial Pars Intercerebralis. The PI is a predominantly peptidergic domain, and neurons in this region are poised to have influence over extended durations[50]. Recent work has implicated a relationship between brainwide behavior-related activity and metabolism[27]. Our observation that neurons involved in regulating metabolism are also modulated by running, albeit in a manner distinct from most other neurons, suggests that the causality of this relationship may be bidirectional.

Why does locomotor behavior have privileged access to virtually all neurons in the fly brain? Neurons in multiple neural pathways would likely benefit from knowledge of current behavior[64]. This activity may modulate ongoing behavior, recapitulate past, or even predict future behavioral action. In artificial intelligence, the utility of proprioceptive feedback to higher-order networks has been demonstrated—in artificial agents trained to solve a variety of tasks, subnetworks charged with representing abstract quantities such as value benefit from knowledge of the agent's behavior[65,66]. Interestingly, artificial neurons in such subnetworks also tend to have activity correlated with the behavior itself[65]. Therefore, locomotor state may provide a useful behavioral context for other computations throughout the brain and it is perhaps not surprising that it elicits the most prominent activity throughout the brain. In short, it is good to know what you are doing.

## Methods

### Genetics and fly rearing
We imaged female 4–7 day-old flies of the following genotype: w/+; UAS-nls-GCaMP6s/+; nSyb-Gal4/UAS-nls-DsRed. UAS-nls-GCaMP6s was a gift from Barry Dickson. We imaged egg-laying command neurons using the split-GAL4 oviDN-SS1[7]. We imaged Dilp neurons using Dilp5-GAL4, and Dh44 neurons using Dh44-GAL4.

### SCAPE light-sheet imaging
Imaging was performed on a SCAPE 2.0 system[44]. In brief, the laser sheet was directed through an upright mounted 20x/1.0NA water immersion objective. Emitted light from the sample was separated into two channels by an image splitter outfitted with two dichroic filters and the detected red and green channels were recorded side-by-side on the camera chip. The imaging speed for these experiments was between 8 and 12 volumes per second, typically covering a volume of ~$450 \times 340 \times 150 \mu m^3$. Variability in size of visible brain volume determined scan speed. In raw data, the voxel size along two dimensions is isotropic and defined by the camera chip, while voxel size in the third dimension is the step size of scanning. Here, the scan dimension was anterior to posterior. Because the light-sheet accesses the brain from an oblique angle, we orthogonalize the coordinate system before further processing, resulting in a typical voxel size of $1.0 \times 1.4 \times 2.4 \mu m$.

### Mount and preparation
We mounted flies to a customized holder consisting of a 3D-printed holder and a laser-cut stainless-steel headplate. We use a spherical treadmill similar to prior designs[67]. We monitor the behavior of the fly at 70 Hz, illuminated by 750 nm LEDs using a Basler acA780 camera outfitted with a VZM-450i lens (Edmund Optics) and a near-IR longpass filter (Midwest Optical LP780-22.5, Graftek Imaging). Depictions of the preparation made in BioRender (Fig. 1a, b).

Our mounting and dissection procedure was very similar to prior work[67] but with a larger dissected window to accommodate SCAPE (Fig. 1b); all dissections that opened up a window similar to Fig. 1b without damaging the brain were deemed successful. After dissection, flies were tested for robust behavior on the spherical treadmill - we defined robust behavior as exhibiting bouts of walking totaling at least one minute in a 5-minute span. All flies that exhibited robust behavior post-dissection were imaged. Most flies that passed these criteria continued to exhibit robust behavior for many minutes, but we only analyzed data from flies that exhibited bouts of walking totaling at least one minute in the first five minutes of imaging. Imaging continued for up to 30 minutes, terminating when a fly no longer exhibited bouts of walking. The mean experiment duration over all flies included in the analysis was 18.1 minutes.

### Motion correction
To perform image registration of our volumetric imaging dataset, we used the NoRMCorre algorithm[68] augmented with an annealing procedure in which the grid size and the range of permitted local displacements gradually decrease with each iteration. At each step, we computed displacements using the activity-independent DsRed channel and applied the inferred displacements to the GCaMP channel.

## Source extraction and deconvolution

ROIs are defined using watershed segmentation applied to the red channel of a temporally averaged volume, resulting in $1631 \pm 109$ ROIs per animal. After motion correction, most cells have negligible residual motion, but in some data sets a small fraction of cells have motion that is too nonlinear to be addressed with NoRMCorre. To quantify residual motion and eliminate non-stationary cells, we compute the squared coefficient of variation, $CV^2 = \mathrm{Var}[\Delta F/F]/\mathrm{Mean}[F]^2$ from the red channel. Most ROIs (>95%) have $CV^2 << 1$, while some have $CV^2 >> 1$ and are discarded. No cells exhibit $CV^2 \approx 1$ (Fig. S1). This refinement of ROIs yields $1419 \pm 78$ stable, single-cell ROIs per animal.

Although this procedure typically reduces motion artifacts to less than 1 voxel for most cells, we further minimize the impact of residual motion by defining the activity of each cell as the ratio of green and red, $F$ = green/red. We then define baseline ratiometric fluorescence, $F_0$ as the best-fit exponential using least absolute deviation (LAD) regression applied to the derivative of $F$, $dF_t = F_{t+1} - F_t$. Specifically, for each cell, $\hat{a}, \hat{b} = \mathrm{argmin}_{a,b} \sum_t |dF_t - dF_0(t,a,b)|$, where $dF_0(t,a,b) = -(a/b) \exp[-t/b]$. We then define $\Delta F/F = (F - F_0)/F0$, where $F_0 = m + a \exp(-t/b)$ and $m = \min[F]$. LAD regression confers robustness to outliers, and working with the derivative of $F$ confers robustness to long-timescale nonstationarity. We find similar but slightly noisier activity using simple $\Delta F/F$ defined on the green channel alone.

## Anatomical alignment across animals

We create a standardized reference frame by coarsely aligning cell locations across flies. Treating every cell as a point, we align the point sets for each brain to a common reference volume using the Gaussian mixture model method developed here: https://github.com/bing-jian/gmmreg.

## Analysis of behavior

We monitor the movement of the spherical treadmill by measuring the total pixel variance between successive frames from the region containing the ball. This unitless estimate of motion-aided behavior segmentation, is described below. In some datasets, the spherical treadmill was removed after 10 minutes of imaging. Here, we measured pixel variance in an ROI around the fly's legs, which provided a measure of behavior we called flailing, consisting of bouts of rapid leg movements.

We analyze fly behavior both by directly tracking motion of the treadmill (described above) and by tracking eight points on the body of the fly using Deep Graph Pose[48] (DGP; Fig. 1c). We hand-labeled the eight selected points in 1771 frames from 26 videos (50–137 frames per video) using the DeepLabCut (DLC)[69] GUI. We then trained DGP on these frames, which augments the supervised loss of DLC with a semi-supervised loss that incorporates additional, unlabeled frames; we found that this significantly improved the pose estimation, even after post hoc smoothing of the DLC markers.

We further segment discrete behaviors from the DGP markers using a semi-supervised sequence model[49]. We chose to label five salient behaviors commonly observed across all flies: running, front and back grooming, abdomen bending, and a quiescent state. We labeled up to 1000 frames for each of the five behaviors for each of 20 flies using the DeepEthogram GUI[70], resulting in a total of 33,756 hand labels (quiescent = 6250, run = 4950, front groom = 5700, back groom = 5480, abdomen bend = 11,376). We supplemented this small, high-quality set of hand labels with a large, lower-quality set of "weak" labels computed using a simple set of heuristics (see details below).

**Semi-supervised behavioral segmentation.** We train a semi-supervised behavioral segmentation model that classifies the DGP markers into one of the five available behavior classes for each time point. The model's loss function contains three terms: (1) a standard supervised loss that classifies a sparse set of hand labels; (2) a weakly supervised loss that classifies a set of easy-to-compute heuristic labels; and (3) a self-supervised loss that predicts the evolution of the DGP markers. Let $\mathbf{x}_t$ denote the DGP markers at time $t$, and let $\mathbf{y}_t$ denote the one-hot vector encoding the hand labels at time $t$ such that the $k^{\mathrm{th}}$ entry is 1 if behavior $k$ is present, else the entry is 0. We assume that the hand labels are only defined on a subset of time points $\mathcal{T} \subseteq \{1, 2, \ldots T\}$. The cross-entropy loss function then defines the supervised objective ($\mathcal{L}_{\mathrm{super}}$) to optimize:

$$\mathcal{L}_{\mathrm{super}} = \sum_{t \in \mathcal{T}} \mathcal{L}_{\mathrm{xent}}(\mathbf{y}_t, f(\mathbf{x}_t)),$$

where $f()$ denotes the sequence model mapping the DGP markers to behavior labels. We now introduce a set of heuristic labels $\tilde{\mathbf{y}}_t$, defined at each time point. Computing the cross-entropy loss on all time points that do not already have a corresponding hand label defines the heuristic objective:

$$\mathcal{L}_{\mathrm{heur}} = \sum_{t \notin \mathcal{T}} \mathcal{L}_{\mathrm{xent}}(\tilde{\mathbf{y}}_t, f(\mathbf{x}_t)).$$

The self-supervised loss requires the sequence model to predict $\mathbf{x}_{t+1}$ from $\mathbf{x}_t$. To properly do so we now expand the definition of the sequence model $f()$ to include two components: an encoder $e()$, which maps the behavioral features $\mathbf{x}_t$ to an intermediate behavioral embedding $\mathbf{z}_t$; and a linear classifier $c()$ which maps $\mathbf{z}_t$ to the predicted labels ($\hat{\mathbf{y}}_t = c(e(\mathbf{x}_t))$). We can now incorporate the self-supervised loss through the use of a predictor function $p()$, which maps $\mathbf{z}_t$ to $\mathbf{x}_{t+1}$, and match $\mathbf{x}_{t+1}$ to the true behavioral features $p(e(\mathbf{x}_{t+1}))$ through a mean square error loss $\mathcal{L}_{\mathrm{MSE}}$ computed on all time points:

$$\mathcal{L}_{\mathrm{pred}} = \sum_{t=1}^{T-1} \mathcal{L}_{\mathrm{MSE}}(\mathbf{x}_{t+1}, p(e(\mathbf{x}_t))).$$

Finally, we combine all terms into the full semi-supervised loss function:

$$\mathcal{L}_{\mathrm{semi}} = \lambda_s \mathcal{L}_{\mathrm{super}} + \lambda_h \mathcal{L}_{\mathrm{heur}} + \lambda_p \mathcal{L}_{\mathrm{pred}},$$

where the $\lambda$ terms are hyperparameters that control the contributions of their respective losses. Note that setting $\lambda_h = \lambda_p = 0$ results in a fully supervised model, while $\lambda_s = \lambda_h = 0$ results in a fully unsupervised model.

For the encoder and predictor networks $e()$ and $p()$ we use a dilated Temporal Convolutional Network (dTCN)[71], which has shown good performance across a range of sequence modeling tasks. Both networks use a two-layer dTCN with a filter size of 9 time steps and 32 channels for each layer, with leaky ReLU activation functions, and weight dropout with probability $p = 0.1$. We use 10 fly videos for training and 10 for testing. All models are trained with the Adam optimizer using an initial learning rate of 1$e$-4 and a batch size of 2000 time points. For the training flies, 80% of frames are used for training, 20% for validation. Training terminates once the loss on validation data begins to rise for 20 consecutive epochs; the epoch with the lowest validation loss is used for testing. To evaluate the models, we compute the F1 score - the geometric mean of precision and accuracy - on the hand labels of the 10 held-out test flies. We average the F1 score over all behaviors and choose the hyperparameters $\lambda_h$ and $\lambda_p$ based on the highest score. We then retrain the model with those hyperparameter settings using all 20 flies to arrive at our final segmentation model. We also performed a small hyperparameter search across the number of layers, channels per layer, filter size, and learning rate, and found that our results are robust across different settings (data not shown).

To construct an ethogram of behavioral state, we use the argmax of predicted behavioral state labels ($\hat{\mathbf{y}}_t$) at every time point. Time points in which $\max[\hat{\mathbf{y}}_t] < 0.75$ are labeled as "undefined" in the ethogram.

**Heuristic labels.** The addition of a large set of easily computed heuristic labels improves the accuracy of the behavioral segmentation[49]. Below, we provide more detail on these heuristics. Note that we choose conservative values for the thresholds in order to decrease the prevalence of false positives. A consequence of this choice is that some time points are not assigned a heuristic label; nevertheless this procedure adds enough high-quality information to substantially improve the models.

*Run.* We first estimate the time points at which a fly is running by utilizing the treadmill motion energy (ME). We transform the treadmill ME to lie in the range [0, 1], then assign the 'run' label to time points when the treadmill ME is above a threshold (0.5).

*Quiescent.* We compute the average ME over all DGP markers for each time point, then denoise this one-dimensional signal with a total variation smoother (the denoise_tv_chambolle filter from the sklearn[72] Python package). We then transform this signal to approximately lie in the range [0, 1] (the 99th percentile is mapped to 1 in order to make this process robust to outliers). We assign the 'quiescent' label to time points when this signal is below a threshold (0.02) and the fly is not running (according to the previous heuristic).

*Abdomen bend.* We compute the average ME over the abdomen markers, then denoise this signal and transform it to approximately lie in the range [0, 1]. We assign the 'abdomen bend' label to time points when this signal is above a threshold (0.9) and the fly is not still or running according to the previous heuristics.

*Front and back groom.* We compute the average ME over the forelimb markers, then denoise this signal and transform it to approximately lie in the range [0, 1]. We assign the 'front groom' label to time points when this signal is above a threshold (0.05), the corresponding back groom signal (computed from the hindlimb markers) is below a threshold (0.02), and the fly is not still, running, or bending its abdomen according to the previous heuristics. We assign the 'back groom' label in an analogous manner.

## Regression model

We regressed each neuron's activity against all behavioral states ($B$={running, front grooming, back grooming, flailing}) filtered using a fitted time constant ($\tau_i$) and temporal shift ($\phi_i$) unique for each cell ($i$). To reflect moments of uncertainty in a fly's behavioral state, we used the behavioral state probabilities ($\hat{y}_{bt}$) rather than the binary behavioral states ($\text{argmax}[\hat{y}_{bt}]$) as regressors. Thus, we model the activity $f$ of cell $i$ at time $t$ as

$$f_{it} \sim \sum_{j=0}^{2} \alpha_{ij} t^j + \sum_{b \in B} \gamma_{bb} \hat{y}_{bti}, \text{ where } \hat{y}_{bti} = b_t \circledast \kappa_{\tau_i \phi_i}.$$

We fit all parameters simultaneously using Sequential Least Squares Quadratic Programming. The $\gamma$ coefficients describe the relative importance of each behavior in accounting for the activity of each cell, while the $\alpha$ coefficients capture drift independent of behavior. The convolution kernel is $\kappa_{\tau_i \phi_i} = (2\tau_i)^{-1} \exp[-(|t - \phi|)/\tau_i]$. This symmetric kernel avoids presuming a causal direction between behavior and neural activity. A cell with a broad kernel should have $|\phi| \geq \tau$, with the sign of $\phi$ determining the direction of potential causality (neural activity that precedes behavior may or may not be causal to the behavior, but neural activity that follows behavior cannot be causal). A lag of $|\phi| \approx \tau$ should not be interpreted as a true lag, but rather a reflection of putative causality with smoothness constraints.

The alternative regression model used the principal components of DGP marker position. Specifically, we used the principal components of the normalized and mean subtracted x and y coordinates of all 8 tracked points. This set of 16 orthogonal regressors were then fed into the same regression model described above in place of the behavioral states.

To test the significance of the fit of each cell by either regression model, we compared variance explained to that from a model that used behavior regressors that were randomly shifted in time. Specifically, we randomly shifted all regressors in time by the same fraction of the total experiment duration, ranging from 33% to 66%, with time points shifted past the end of the experiment wrapping around to the beginning. We generated five instances of this shifted fit and required that the original fit produced larger $r^2$ than all of them. The regression fit for cells that failed this test were treated as not significant, regardless of their $r^2$ value.

## Significance of residual dynamics

To ascertain the degree of temporal structure in the residual activity after subtracting the regression model fit, we performed a Ljung-Box test of autocorrelation in the residual dynamics. For every cell, we performed this test on all lags between 10 and 610 frames (approximately corresponding to 1 second and 1 minute, respectively). Every lag and every cell from every fly yielded a p-value lower than $10^{-10}$.

## Dimensionality reduction

We performed PCA on the residual activity after subtracting the regression model fit. We quantified the dimensionality of this residual activity as the number of PCA modes that maximize the log-likelihood of the lower dimensional subspace on held-out data. We fit the principal components on 80% of all time points and evaluate the log-likelihood on the remaining 20%.

To quantify the degree of approximate sparseness of PCA modes without selecting a threshold, we calculate the participation ratio of each principal component vector $\vec{v}_j$ as

$$S_j = \frac{\left(\sum_k v_{jk}^2\right)^2}{\sum_k \left(v_{jk}^4\right)}.$$

Intuitively, this gives an estimate of how many elements of each mode are large (significantly nonzero), without having to choose an arbitrary threshold. The participation ratio of a zero-mean Gaussian vector is ~0.33, which is a useful null hypothesis for the existence of either sparse or dense structure in the PCA modes. We define the number of active neurons ($n$) in each mode as sparseness ($S$) multiplied by the total number of neurons.

We sorted residual activity by behavior label and then performed PCA separately on each behavior's set of time points to quantify the residual subspace ($\mathbf{X}_b$) of each behavior $b$. To compare the subspaces of two behaviors, for example running and the quiescent state, we quantified the common variance explained and the common dimensionality. We defined common variance explained ($E_{mb}$) for $m$ modes as

$$E_{mb} = \frac{\sum_{j=0}^{m} \mathbf{X}_{b_0} \vec{v}_{jb}}{\sum_{j=0}^{m} \lambda_{jb}},$$

where $b$ is the behavior on which the PCA modes were defined, and $b_0$ is the other behavior. Similarly, we define common dimensionality by cross validating the projection of one subspace onto the modes of the other ($\mathbf{X}_{b_0} \vec{v}_{jb}$). See 'Spatial Organization' section for explanation of calculating intra-mode distances and spatial organization.

## Clustering

We performed agglomerative hierarchical clustering on residual neural activity using Euclidean affinity and ward linkage.

To look for structure in the data at all spatial scales without defining arbitrary parameters for the number of expected clusters or an affinity threshold, we identified significant clusters using cross-validation. We performed clustering on 80% of the time points, and evaluated the validity of the identified clusters on the remaining 20%. Specifically, we evaluated the intra-cluster variance on held-out time points for each cluster and for size-matched samples from its parent cluster. The number of selected samples was

$$N_{samples} = \min\left[\binom{N_p}{N_c}, 100\right],$$

where $N_p$ and $N_c$ are the number of neurons in the parent and child cluster, respectively. A child cluster was deemed significant if its test variance was less than that of the samples ($p < 0.05$). Both a child and its parent cluster can be significant. See 'Spatial Organization' section for explanation of calculating intra-cluster distances and cluster organization.

## Spatial organization

We analyzed the spatial organization of sparse binary patterns. For our analysis of cluster organization, these patterns directly corresponded to cluster labels. For the corresponding analysis of organization of PCA modes, an intermediate binarization step was required. To approximate each PCA mode as a binary pattern, we set large outliers (greater than five standard deviations from the mean) in the original mode to 1 and all other cells to 0.

We defined the Euclidean distance for each binary pattern by first reflecting the brain along the midline—thus, the lateral coordinate of each cell was equal to its distance from the midline (Fig. S6F). We then compute the Euclidean distance between the coordinates of each cell in a binary pattern. We performed this analysis on both the identified and randomly shuffled patterns of the same size to validate our results.

## Reporting summary

Further information on research design is available in the Nature Portfolio Reporting Summary linked to this article.

## Data availability

Source data are provided with this paper. The datasets generated during the current study are publicly available in NWB format in a Figshare database at https://doi.org/10.6084/m9.figshare.23749074. An accompanying source data file is named "datasets_for_each_figure.xlsx".

## Code availability

Data was collected using custom Matlab software (https://github.com/schafferEvan/VIP) interfacing with Andor acquisition system (andor.oxinst.com). Analyses were performed using custom Python and Matlab code that can be found in the following Github repositiories (package versions are specified in the respective requirements.txt files in each repository): Behavioral data processing: https://github.com/thematthethatt/daart (DOI: 10.5281/zenodo.8277452), Neural data processing: https://github.com/schafferEvan/VIP (ref. 73, DOI: 10.5281/zenodo.8263548), Analysis: https://github.com/schafferEvan/flygenvectors (ref. 74, DOI: 10.5281/zenodo.8263524).

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

## Acknowledgements

We would like to thank Tanya Tabachnik for the design and manufacturing of our fly treadmill, Barry Dickson for generously sharing fly stocks, Armaan Ahmed, Virginia Devi-Chou, and Benjamin Lucero for assistance in processing data, and members of the Axel, Hillman, and Paninski labs for helpful comments and suggestions. This work was supported by grants from the Simons Foundation (481778, E.S.S.; 542951, L.F.A., E.M.C.H., R.A.; 543023, L.P.), the NSF Graduate Research Fellowship Program DGE 16-44869 (N.M.), BRAIN Initiative Awards UF1NS108213 and U01NS094296 (E.M.C.H.), the NSF NeuroNex Award DBI-1707398 (L.F.A., L.P.), the Gatsby Charitable Foundation (L.F.A.), and the Howard Hughes Medical Institute (R.A.).

## Author contributions

Conceptualization, E.S.S., N.M., M.R.W., W.L., L.P., L.F.A., E.M.C.H., R.A. Methodology, N.M., E.S.S., M.R.W., W.L., J.F., L.P., L.F.A., E.M.C.H. Validation—Iterations of Fly Imaging, E.S.S., N.M., W.L., J.F. Software, N.M., E.S.S., M.R.W., W.L., K.P., V.V., E.M.C.H. Formal Analysis, E.S.S., N.M., M.R.W. Investigation, N.M., E.S.S., W.L., J.F., M.V. Data Management, E.S.S., N.M., M.V. Software for Data Curation, N.M., E.S.S., M.R.W. Writing —original draft, E.S.S. Writing—Preparation of Manuscript, N.M, E.S.S. Writing—review & editing, E.S.S., N.M., M.R.W., W.L., L.P., L.F.A., E.M.C.H., R.A. Visualization, E.S.S. Supervision, R.A., E.M.C.H., L.F.A., L.P. Funding acquisition, N.M., E.S.S., L.P., L.F.A., E.M.C.H., R.A.

## Competing interests

The authors declare no competing interests.
