## [Peer Review File · Nature Communications]

The spatial and temporal structure of neural activity across the fly brainEditorial Note: This manuscript has been previously reviewed at another journal that is not operating a transparent peer review scheme. This document only contains reviewer comments and rebuttal letters for versions considered at *Nature Communications*.

REVIEWER COMMENTS

Reviewer #1 (Remarks to the Author):

Summary: This manuscript aims to approach an ambitious question: how is behavior distributed in brain-wide circuits. With the fast expansion of experimental methods, the authors note that neuroscience is at a precipice where a local understanding of brain-behavior is insufficient. This work pushes in the direction of these big-picture questions using state-of-the-art tools (SCAPE imaging and behavioral analysis) to take a step towards this direction. As prior reviewers noted, there are clear limitations even to the impressive modern-day technology in the scope of circuitry that can be imaged and analyzed. Regardless I find that this work does make significant progress in this direction. There are a few places that I feel additional quantitative rigor would help drive the main points, which I have outlined below. In addition there are some more minor points (including an oddity in the definition of sparsity) that I feel should be addressed. With appropriate additions this will be a nice addition to the literature.

Major Comments:

- A clear strength of the work is in the analysis of the residuals. I believe this is a vital aspect of validating the analysis. Within this, however, I think the authors can be much more quantitative and include metrics on the residual, such as measures of independence (the Ljung-Box Quantile Test for example).
- One potential weakness is the exclusion of the temporal relationship of the data. In essence it seems that each time-point is treated independently in the dimensionality reduction. The claim of context-dependent activity, however, would be much stronger and clearer if not isolated from the dynamics of the circuit. In particular, middling correlations can be due to temporally local fluctuations, or long-term changes in brain state. I wonder if the authors think that such an approach is possible given the extended dynamics of nuclear calcium or if the extended time-scale would confound such an analysis despite deconvolution.
- Another weakness is that the correlative aspect of this work might be a bit misleading about the causal role of the relationship between individual neurons and behavior. Directly correlating neurons to all behaviors without accounting for the internal correlation structure of the behavior is tricky. For example in general, least-squares under noisy conditions will always return non-zero values for all coefficients i.e., all behaviors are related to all neurons. It's not clear how this was controlled for in the regression model analysis.
- Motion artifacts: Motion correction is mentioned, and I can see the prior discussion between the prior reviewers and authors. I appreciate the efforts the reviewers went through to quantify motion, however it might not take much motion, when averaged over many ROIs, to add up to a noticeable effect. I

wonder if one option to further demonstrate that the motion correction worked would be to see if per-joint motion is decodable in addition to overall motion levels. I bring this up simply because the overall motion (sum-of-absolute values) seems to correlate to behavioral state as per the plots shown. Decoding joints would show that the information is not just “animal moves more” but is more specific and thus less correlated to a major potential cause of motion.

Minor Comments:

- There is a reference to causality as a time shift: Time shifts do not equal causality! (e.g., with respect to tau) This might trigger people...
- In the methods, it is not clear how tau is fit in the regression model. Is this part of the regression or is tau fit separately in some way?
- The sparsity metric is a bit confusing to me. Why is S defined as a metric on v^2 ? Usually sparsity is approximated as the ratio of the squared l1 to the squared l2 norm: $S = \frac{\|v\|_1^2}{\|v\|_2^2}$ (and some people even omit the square, i.e. $S = \frac{\|v\|_1}{\|v\|_2}$). Squaring the values will likely inflate (i.e., deflate) the sparsity. It should also clearly be stated how the “# active neurons” is calculated. It looks like $\text{sparsity} \times (\text{total \# neurons})$, but this should be explicit.
- The introduction is a bit redundant: some phrases are repeated without adding more to the conceptual development
- The authors use the claim that the dimensionality of the body motion is on par with the number of behavioral states. This is a bit confusing as it does not seem to need to be the case. Some behaviors can have high variance activity, and others might share activity patterns with other states, meaning that the measure of behavior could in theory be quite independent of body motion dimensionality.

Reviewer #2 (Remarks to the Author):

In this manuscript, the authors use SCAPE microscopy to record the calcium activity of a large number of neurons in the dorsal part of the fly brain, while the fly performs a range of spontaneous behaviors. The authors use a suite of techniques to analyze this large dataset - they use regression analysis to identify neural activities that are directly correlated with the behavioral state of the fly, principal component analysis and hierarchical clustering to identify structure in the neural activity. Using these analyses, the authors broadly conclude that: 1) the activity of a large group of neurons is directly correlated with global behavioral states like walking or flailing and less correlated with localized behaviors like grooming; 2) neurons with similar behavioral time constants were clustered spatially; 3) residual neural activity, i.e., neural activity not linked to behavioral state, was complex and high dimensional, yet it was also observed that neurons whose activities were correlated were also spatially clustered or anatomically linked (like bilateral symmetry).

The methodology used, the dataset obtained and the analyses used by the authors are impressive. These results clearly show that behaviors like walking and running are present as neural activity

throughout the whole brain, which has not been shown before at a single-cell resolution. The analyses are sound, and the authors addressed most concerns by other reviewers properly.

However, to obtain single-neuron activity, the authors had to make a trade-off with the nls-GCaMP. The authors say, “Imaging nuclear calcium precluded seeing very fast dynamics.”, which is an understatement of the issue. They also say, “We reasoned that any increase in low-pass filtering introduced by using a nuclear localized indicator was greatly offset by the advantage of allowing cellular-level spatial resolution.” But it is not clear if the nucleus's calcium activity is indeed a simple low-pass filtering of the electrical activity of the neuron. Even if it is a simple low-pass filtering, it may not be consistent across all neurons. Further, it is not clear if it can be used to say anything about circuit dynamics in the brain. (As an analogy, it is like saying that liver glucose production is correlated to the visual processing in the brain.) There may be some correlation, of course, but they are too far away (as authors know, unlike mammalian neurons, the cell bodies of fly neurons are outside the brain, far from where electrical activity happens.) and nucleus calcium activity is vastly different in time constants spanning 1s to 40s, as the results show, which is simply too long to correlate the electrical activity of neurons. Thus, calling the nucleus calcium activity ‘neural activity’, which usually means electrical activity, seems to stretch the term too much. Although a different term, such as ‘neural metabolism’, may represent the results better, I strongly suggest that, at a minimum, authors should state potential pitfalls in the interpretation of the results from using nls-GCaMP. Or they should cite a study showing the relationship between nucleus calcium activity and electrical neural activity in fly neurons.

Reviewer #3 (Remarks to the Author):

In this study, Schaffer, Mishra et al. present data to help bridge single neuron to large-scale brain activity studies. They confirm that behavior is represented broadly in the brain and show that most neurons are involved rather than a few neurons with broad projections. This approach is very complementary to previous studies recording from the neuropil in flies. Furthermore, an analysis of residual activity extracted units with similar statistics as cell type extracted from molecular and anatomical methods. These results contextualize traditional studies of small circuits involved in specific functions in flies and reaffirm the global broadcast of behavior information in the brain found across animals. The data is quantified thoughtfully (e.g. using rate of variance explained, statistically defined clusters), and overall the study is explained clearly.

Major points:

Line 202: “residual activity shows no obvious relationship to behavioral state.” Some obvious relationship would be a correlation with onset or offset of the behavior, and some signals in Fig. 4A and B seem consistent with this. It would be great if the authors could add some analysis or comments on this point.

Although the analysis is already extensive, only a binary classification of behavior and correlative/linear methods have been used, so much more insight could be gained from this dataset in future studies.

Furthermore, this dataset would be very helpful for teaching purposes. The significance of the study would thus be greatly increased by broadly sharing the data.

Minor points:

The behavior analysis is complex and employs many different tools. Please consider adding a flow diagram to help the reader.

It would be helpful to the reader to see how the recorded area relate to the whole brain when introducing the data. For example the authors could show the data on a template brain (like S2 F,G) earlier (in Fig 1 or S1).

Line 56: "This provides a comprehensive picture of activity in the fly brain." This is misleading as the number of neurons is more than one order of magnitude smaller than the number of neurons in the brain.

Line 71: Please clarify here or in the methods whether voxel size (defined by the optics) is the same as spatial sampling rate.

Fig2: The organization of this figure is confusing. Please consider re-organizing it to have all time constant together.

Line 120: "Notably, this biologically meaningful heterogeneity in adjacent neurons would be masked by imaging the neuropil formed by these neurons." I am not sure whether this is accurate as adjacent cell bodies could correspond to projections in different neuropil areas.

Line 253: The example provided don't reflect the scale down to a couple of neurons. Consider adding an example with just one or two neurons per neuron type.

Line 281: "Thus, most or perhaps all neurons that exhibit a negative correlation in panneuronal data are the peptidergic cell type Dilp." Maybe add "in the dorsal brain" to clarify that there could be negatively correlated neurons elsewhere.

Fig5A: $\int_0^{2\pi} \int_0^{\pi} \sin^2 \theta d\theta = \int_0^{2\pi} \int_0^{\pi} \sin^2 n\theta d\theta$, so changing the frequency is maybe not the best way to convey the change that would be detected by the variance analysis. Please consider changing the amplitude of the oscillations to better illustrate what is detected in the rest of the figure.

Dilp and Dh44: A quantification of the comparison between the clusters found from large scale recording and the Dilp and Dh44 would help make the point that they are the same. For example is it possible to quantify the likelihood of these neuron types to be one of the clusters detected from the functional analysis, based on their position and number of neurons?

Line 366: Please give more details on the dissection process. Are there exclusion criteria for flies that were damaged after surgery?

Line 369: Please provide more information on the length of experiments.

Fig. 6, S6 and line 191, 247, 490: "folding the brain along the midline" this is confusing as it would result in flipping the z axis for one half of the brain.

REVIEWER COMMENTS

Reviewer #1 (Remarks to the Author):

Summary: This manuscript aims to approach an ambitious question: how is behavior distributed in brain-wide circuits. With the fast expansion of experimental methods, the authors note that neuroscience is at a precipice where a local understanding of brain-behavior is insufficient. This work pushes in the direction of these big-picture questions using state-of-the-art tools (SCAPE imaging and behavioral analysis) to take a step towards this direction. As prior reviewers noted, there are clear limitations even to the impressive modern-day technology in the scope of circuitry that can be imaged and analyzed. Regardless I find that this work does make significant progress in this direction. There are a few places that I feel additional quantitative rigor would help drive the main points, which I have outlined below. In addition there are some more minor points (including an oddity in the definition of sparsity) that I feel should be addressed. With appropriate additions this will be a nice addition to the literature.

Major Comments:

- A clear strength of the work is in the analysis of the residuals. I believe this is a vital aspect of validating the analysis. Within this, however, I think the authors can be much more quantitative and include metrics on the residual, such as measures of independence (the Ljung-Box Quantile Test for example).

We appreciate this suggestion, and we have incorporated a quantitative assessment at the beginning of our analyses of the residual dynamics. Using a Ljung-Box Quantile Test, we found that every neuron in our dataset exhibited significant residual correlations on multiple timescales. This is consistent with and in many ways strengthens our observations that dimensionality reduction and clustering reveal richness in these dynamics. We have added the following to this section of the Results:

“After large-scale locomotion- and other behavior-related activity has been regressed out, the residual activity exhibits rich dynamics across both space and time (Fig. 4A), with all neurons exhibiting significant residual dynamics across timescales, from seconds to minutes ($p < 1e-10$, Ljung-Box test. See Methods).”

The corresponding section in the Methods, “Significance of residual dynamics,” reads: “To ascertain the degree of temporal structure in the residual activity after subtracting the regression model fit, we performed a Ljung-Box test of autocorrelation in the residual dynamics. For every cell, we performed this test on all lags between 10 and 610 frames (approximately corresponding to 1 second and 1 minute, respectively). Every lag and every cell from every fly yielded a p-value lower than 10^{-10} .”

- One potential weakness is the exclusion of the temporal relationship of the data. In essence it seems that each time-point is treated independently in the dimensionality reduction. The claim of context-dependent activity, however, would be much stronger and clearer if not isolated from the dynamics of the circuit. In particular, middling correlations can be due to temporally local

fluctuations, or long-term changes in brain state. I wonder if the authors think that such an approach is possible given the extended dynamics of nuclear calcium or if the extended time-scale would confound such an analysis despite deconvolution.

We would like to thank the reviewer for identifying this potentially confusing point. PCA identifies the patterns that explain the most variance in a dataset, and each such pattern (a principal component or “mode”) is defined by a spatial pattern and a temporal pattern. The temporal pattern of the first mode, for example, describes the dynamics that capture the most variance, while the corresponding spatial pattern indicates how much that mode contributes to explaining the activity of each cell. In our manuscript, we have focused on the spatial patterns because they show interesting organization. However, we recognize that our omission of the temporal patterns may have led to confusion. For completeness, we have added the temporal patterns corresponding to the modes shown at the top of Figure 4G-I to Figure S4. The main text reference to this now reads:

“Each PCA mode is sparse and therefore dominated by the activity of a small group of neurons with idiosyncratic yet similar dynamics (Fig. S4).”

The new panel in Fig. S4 and corresponding legend is:

“(A) Timecourse of each of the three modes shown at the top of Figs. 4G-I, respectively, with ethograms shown below. Dynamics are in arbitrary units.”

- Another weakness is that the correlative aspect of this work might be a bit misleading about the causal role of the relationship between individual neurons and behavior. Directly correlating neurons to all behaviors without accounting for the internal correlation structure of the behavior is tricky. For example in general, least-squares under noisy conditions will always return non-zeros values for all coefficients i.e., all behaviors are related to all neurons. It’s not clear how this was controlled for in the regression model analysis.

The reviewer raises a very important point that we addressed in our analysis but neglected to make clear in our manuscript. As the reviewer points out, one expects non-zero values for all regression coefficients, and this could lead to the mistaken conclusion that all neurons encode all behaviors. To control for these spurious correlations, we repeatedly fit every cell using behavior regressors that were randomly shifted in time. If a cell is fit equally well by this model with shifted regressors, it implies that the activity of that cell is not meaningfully related to behavior. We therefore treated the regression fit for such cells as not significant, regardless of how high the correlation values were. We have added the following sentence to the regression section of the Results:

“We assessed the significance of the fit to each cell by randomly shifting regressors in time (Methods).”

And the corresponding section of the Methods now reads:

“To test the significance of the fit of each cell by either regression model, we compared variance explained to that from a model that used behavior regressors that were randomly shifted in time. Specifically, we randomly shifted all regressors in time by the same fraction of the total experiment duration, ranging from 33% to 66%, with timepoints shifted past the end of the experiment wrapping around to the beginning. We generated 5 instances of this shifted fit and required that the original fit produced larger r^2 than all of them. The regression fit for cells that failed this test were treated as not significant, regardless of their r^2 value.”

- Motion artifacts: Motion correction is mentioned, and I can see the prior discussion between the prior reviewers and authors. I appreciate the efforts the reviewers went through to quantify motion, however it might not take much motion, when averaged over many ROIs, to add up to a noticeable effect. I wonder if one option to further demonstrate that the motion correction worked would be to see if per-joint motion is decodable in addition to overall motion levels. I bring this up simply because the overall motion (sum-of-absolute values) seems to correlate to behavioral state as per the plots shown. Decoding joints would show that the information is not just “animal moves more” but is more specific and thus less correlated to a major potential cause of motion.

We share the reviewer’s recognition that this is an extremely important step in our analyses, and we appreciate your efforts to think of additional ways to test for possible confounds. We have given careful thought to the proposed decoding analysis, and we are concerned that the conclusions from such a test would be ambiguous. Specifically, both a success or a failure in decoding of joint position could hypothetically result from a success or a failure of motion correction. Consider the following simple example: If the brain were to move every time the fly moved one leg, failed motion correction would result in successful decoding. Conversely, if neurons were active when the fly moved that leg, but motion of the brain were uncorrelated with locomotion, failed motion correction would result in failed decoding. Thus, failure to decode could either indicate that small artifacts induced by brain motion obscured the signal or that the neurons we imaged do not encode joint position. Similarly, successful decoding could either indicate that the neurons we imaged encode joint position or that joint position can be decoded from minute artifacts in a small number of neurons. We have taken this opportunity to expand and clarify our detailed examination of brain motion before and after

motion-correction and the relationship between brain motion and behavioral state, as described below.

To the reviewer’s concern that “the overall motion (sum-of-absolute values) seems to correlate to behavioral state as per the plots shown,” we wish to point out that although there may appear to be a correlation, this is actually not the case, and we have attempted to make this easier to see by eye. The correlation coefficients between brain motion and the running state in the example traces in Fig. S1C are -0.06 and -0.02 (raw and registered, respectively), which we have added to the figure legend. Epochs of large brain motion occur with similar frequency and magnitude during running and quiescence, examples of which have been added on the bottom of Fig. S1C. This lack of correlation holds across all flies - the average correlations between brain motion and the running state are 0.02 and 0.04 (raw and registered, respectively. Fig. S1E).

To the reviewer’s concern that “it might not take much motion, when averaged over many ROIs, to add up to a noticeable effect,” we wish to point out that we observe neural activity that correlates with running in the majority of individual neuron traces, which could not be due to an effect of averaging over many ROIs. Moreover, as described above, movement of the brain is not correlated with running, and thus even if we were to skip motion correction entirely, this would manifest as added noise, not a spurious correlation with running. To clarify the degree of brain motion we observe before and after registration, we have added representative example images to Figure S1. As can be seen in Fig. S1D (reproduced below), during an epoch in which the raw brain stability score is low (point ‘x’), the detailed alignment of individual cells after registration is still nearly perfect. Consistent with this, the stability score of the registered trace in Fig. S1C is 0.998, and the average registered stability is 0.997 (Fig. S1E). The new parts of Figure S1 are below, followed by the corresponding section of the figure legend:

“(C) Top, motion of the brain volume before (dark red) and after (light red) registration, quantified as the correlation coefficient between red fluorescence and a single template image, with behavioral state shown below (same color code as in 'A'). The correlation between running and

the raw and registered traces are -0.06 and -0.02, respectively. *Bottom*, magnified view of two epochs in which the brain moves substantially, during either running (left) or quiescence (right), with time points shown in 'D' indicated by 'x' and 'y'. (D) *Top*, snapshots of dsRed fluorescence from timepoints indicated by 'x' and 'y' markers in 'C', which are during and after an epoch of brain motion, respectively. *Middle*, magnified view of regions indicated by a cyan box above. *Bottom*, the same magnified regions after motion correction.”

Minor Comments:

- There is a reference to causality as a time shift: Time shifts do not equal causality! (e.g., with respect to tau) This might trigger people...

We agree with the reviewer - we used time shifts in our analysis to permit causal and acausal interactions, but we did not intend to imply that a time shift was proof of causality. We have updated this language to reflect this point. The sentence in the Results now reads: “We allowed for both potentially causal and acausal relationships between behavior and neural activity using a cell-specific temporal shift (ϕ_i) of neural activity relative to the annotated behaviors (Methods).”

The corresponding sentences in the Methods now read:

“This symmetric kernel avoids presuming a causal direction between behavior and neural activity. A cell with a broad kernel should have $|\phi| \geq \tau$, with the sign of ϕ determining the direction of potential causality (neural activity that precedes behavior may or may not be causal to the behavior, but neural activity that follows behavior cannot be causal). A lag of $|\phi| \approx \tau$ should not be interpreted as a true lag, but rather a reflection of putative causality with smoothness constraints.”

- In the methods, it is not clear how tau is fit in the regression model. Is this part of the regression or is tau fit separately in some way?

We have added the following line to the regression section of the Methods: “We fit all parameters simultaneously using Sequential Least Squares Quadratic Programming.”

- The sparsity metric is a bit confusing to me. Why is S defined as a metric on v^2 ? Usually sparsity is approximated as the ratio of the squared l1 to the squared l2 norm: $S = \|v\|_1^2 / \|v\|_2^2$ (and some people even omit the square, i.e. $S = \|v\|_1 / \|v\|_2$). Squaring the values will likely inflate (i.e., deflate) the sparsity. It should also clearly be stated how the “# active neurons” is calculated. It looks like $\text{sparsity} \times (\text{total \# neurons})$, but this should be explicit.

The reviewer is correct in how the number of ‘active neurons’ is calculated, and we have added the following sentence to the Methods:

“We define the number of active neurons (n) in each mode as sparseness (S) multiplied by the total number of neurons.”

As the reviewer notes, the choice between these three definitions of sparseness is somewhat subjective, but we defined S as a metric on v^2 because we feel it best aligns with the intuition of how one wants a sparseness metric to behave. For example, if you let v be an array of length 100 with elements drawn from a Gaussian with zero mean and variance 0.01,

but you add 10 to 4 of the elements, our metric would say that the number of active neurons $n=4.02$, but the two alternative metrics give $n=5.66$ and $n=2.38$, respectively. Our metric best captures that 4 elements in the array are much larger than the others.

- The introduction is a bit redundant: some phrases are repeated without adding more to the conceptual development

We have revised the entire introduction to eliminate redundancy and improve clarity.

- The authors use the claim that the dimensionality of the body motion is on par with the number of behavioral states. This is a bit confusing as it does not seem to need to be the case. Some behaviors can have high variance activity, and others might share activity patterns with other states, meaning that the measure of behavior could in theory be quite independent of body motion dimensionality.

The reviewer is correct. We overemphasized the quantitative agreement between the number of states and the dimensionality of movement - it is true that these quantities did not need to match but satisfying that they did. Our point is simply that movement itself is low dimensional, which helps to explain why the regression onto neural data did not improve when we used raw movement as regressors rather than behavioral states. Our sentence in the Results section on this point was unnecessary, and we have removed it. We have also updated the legend to Figure S2E to say:

“(E) Variance of DGP marker position explained by each principal component (DGP Mode). Average dimensionality of raw behavior as calculated by participation ratio is 5.0, consistent with the observation that the 'markers' model in 'A-D' performs similarly to the 'states' model.”

Reviewer #2 (Remarks to the Author):

In this manuscript, the authors use SCAPE microscopy to record the calcium activity of a large number of neurons in the dorsal part of the fly brain, while the fly performs a range of spontaneous behaviors. The authors use a suite of techniques to analyze this large dataset - they use regression analysis to identify neural activities that are directly correlated with the behavioral state of the fly, principal component analysis and hierarchical clustering to identify structure in the neural activity. Using these analyses, the authors broadly conclude that: 1) the activity of a large group of neurons is directly correlated with global behavioral states like walking or flailing and less correlated with localized behaviors like grooming; 2) neurons with similar behavioral time constants were clustered spatially; 3) residual neural activity, i.e., neural activity not linked to behavioral state, was complex and high dimensional, yet it was also observed that neurons whose activities were correlated were also spatially clustered or anatomically linked (like bilateral symmetry).

The methodology used, the dataset obtained and the analyses used by the authors are impressive. These results clearly show that behaviors like walking and running are present as neural activity throughout the whole brain, which has not been shown before at a single-cell resolution. The analyses are sound, and the authors addressed most concerns by other reviewers properly.

However, to obtain single-neuron activity, the authors had to make a trade-off with the nls-GCaMP. The authors say, “Imaging nuclear calcium precluded seeing very fast dynamics.”, which is an understatement of the issue. They also say, “We reasoned that any increase in low-pass filtering introduced by using a nuclear localized indicator was greatly offset by the advantage of allowing cellular-level spatial resolution.” But it is not clear if the nucleus's calcium activity is indeed a simple low-pass filtering of the electrical activity of the neuron. Even if it is a simple low-pass filtering, it may not be consistent across all neurons. Further, it is not clear if it can be used to say anything about circuit dynamics in the brain. (As an analogy, it is like saying that liver glucose production is correlated to the visual processing in the brain.) There may be some correlation, of course, but they are too far away (as authors know, unlike mammalian neurons, the cell bodies of fly neurons are outside the brain, far from where electrical activity happens.) and nucleus calcium activity is vastly different in time constants spanning 1s to 40s, as the results show, which is simply too long to correlate the electrical activity of neurons. Thus, calling the nucleus calcium activity ‘neural activity’, which usually means electrical activity, seems to stretch the term too much. Although a different term, such as ‘neural metabolism’, may represent the results better, I strongly suggest that, at a minimum, authors should state potential pitfalls in the interpretation of the results from using nls-GCaMP. Or they should cite a study showing the relationship between nucleus calcium activity and electrical neural activity in fly neurons.

We have given serious thought to the reviewer’s concerns about NLS-GCaMP, and we have updated our manuscript accordingly.

The reviewer’s chief concern is that nuclear calcium activity might not reflect action potentials, and that therefore “it is not clear if it can be used to say anything about circuit dynamics in the brain.” We wish to point out that we are not the first to report neural activity in *Drosophila* using a nuclear reporter, and previous results support our interpretation. For example, Jung et al. compared the reporter we used (NLS-GCaMP6s) to cytosolic GCaMP6s expressed in P1 follower neurons¹. They observed responses to optogenetic stimulation of P1 neurons with both the nuclear and cytosolic reporter, suggesting that NLS-GCaMP6s can be used as a readout of synaptic input to these neurons.

Similarly, older studies such as Weislogel et al. compared cytosolic and nuclear GCaMP3 responses to footshock and odor either localized to Kenyon Cells or using a sparse panneuronal driver². The cytosolic and nuclear reporters were qualitatively similar in their responses to footshock, but the nuclear reporter did not detect odor responses. These results suggest that older nuclearly localized reporters are also a readout of synaptic input, though potentially less sensitive than cytosolic reporters. This is also consistent with the fact that older versions of GCaMP are much less sensitive than newer ones.

Nuclear-localized GCaMP has also been used widely in other species as a reporter of neural activity, including in other invertebrates such as *C. elegans*. For example, Schrodell et al.

demonstrated very similar stimulus-evoked calcium responses between cytosolic GCaMP5K and nuclear-localized GCaMP5K in chemosensory neurons in *C. elegans*³. Kato et al. further used nuclear-localized GCaMP5K to identify neural activity that predicts behavior in *C. elegans*⁴.

The efficacy of nuclear calcium as an indicator of neural activity is not surprising given the pathways that are conserved across species for trafficking of calcium into the nucleus for activity-dependent gene regulation (See for example Bading⁵). Taken together, evidence from prior work suggests that NLS reporters are valid readouts of neural activity with appropriately considered caveats. We did not adequately explain this in the previous version of our manuscript, so we have added the following to the beginning of the Results:

“Nuclear calcium reporters have been shown to be faithful readouts of neural activity (Weislogel et al., 2013; Jung et al., 2020); they may preclude seeing fast dynamics and small changes in neural activity but offer the substantial benefit of easily resolving individual neurons. We therefore reasoned that any increase in low-pass filtering introduced by using a nuclear localized indicator was greatly offset by the advantage of allowing cellular-level spatial resolution.”

Finally, to the reviewer’s concern that the filtering properties of nuclear imaging “may not be consistent across all neurons,” we wish to point out that this concern could be raised toward any calcium imaging experiment – an inherent caveat in using a fluorescent reporter of calcium as a readout of neural activity is the possibility that the relationship between neural activity and the measurable proxy is not identical for all neurons.

1. Jung, Y. *et al.* Neurons that Function within an Integrator to Promote a Persistent Behavioral State in *Drosophila*. *Neuron* **105**, 322–333.e5 (2020).
2. Weislogel, J.-M. *et al.* Requirement for nuclear calcium signaling in *Drosophila* long-term memory. *Sci. Signal.* **6**, ra33 (2013).
3. Schrödel, T., Prevedel, R., Aumayr, K., Zimmer, M. & Vaziri, A. Brain-wide 3D imaging of neuronal activity in *Caenorhabditis elegans* with sculpted light. *Nat. Methods* **10**, 1013–1020 (2013).
4. Kato, S. *et al.* Global brain dynamics embed the motor command sequence of *Caenorhabditis elegans*. *Cell* **163**, 656–669 (2015).
5. Bading, H. Nuclear calcium signalling in the regulation of brain function. *Nat. Rev. Neurosci.* **14**, 593–608 (2013).

Reviewer #3 (Remarks to the Author):

In this study, Schaffer, Mishra et al. present data to help bridge single neuron to large-scale brain activity studies. They confirm that behavior is represented broadly in the brain and show that most neurons are involved rather than a few neurons with broad projections. This approach is very complementary to previous studies recording from the neuropil in flies. Furthermore, an analysis of residual activity extracted units with similar statistics as cell type extracted from molecular and anatomical methods.

These results contextualize traditional studies of small circuits involved in specific functions in flies and reaffirm the global broadcast of behavior information in the brain found across animals.

The data is quantified thoughtfully (e.g. using rate of variance explained, statistically defined clusters), and overall the study is explained clearly.

Major points:

Line 202: “residual activity shows no obvious relationship to behavioral state.” Some obvious relationship would be a correlation with onset or offset of the behavior, and some signals in Fig. 4A and B seem consistent with this. It would be great if the authors could add some analysis or comments on this point.

Although the analysis is already extensive, only a binary classification of behavior and correlative/linear methods have been used, so much more insight could be gained from this dataset in future studies. Furthermore, this dataset would be very helpful for teaching purposes. The significance of the study would thus be greatly increased by broadly sharing the data.

We will make the dataset publicly available upon publication. We agree with the reviewer that our data are a potentially valuable resource to the community.

We have also incorporated the reviewer’s suggestion to examine the relationship between residual dynamics and transitions between behavioral states. As described below, we compared the average slope of activity of each neuron in the final half second preceding a transition from running to quiescence or vice versa to the activity of that neuron earlier in the corresponding bout of behavior. Consistent with what the reviewer identified by eye in Figures 4A-B, we saw a small but highly significant effect - more neurons had negative changes in activity in transitions from running to quiescence, and more neurons had positive changes in activity in transitions from quiescence to running. The fact that this effect is small could in principle be due to a small but reliable group of neurons that encode these behavioral state transitions. However, we saw the same effect upon shuffling cell identities before computing the slope of activity of each cell, suggesting that no reliable group of transition-encoding neurons exist. Rather, there is just a general bias toward increased (decreased) activity preceding transitions to running (quiescence), and any neuron may exhibit this activity in one behavioral bout and not the next. This suggests that a simple coding model will not capture these dynamics, and a more in depth analysis is beyond the scope of this manuscript. We have added the following text to the Results:

“For example, the residual activity of some cells appears to include dynamics related to transitions between states (Fig. 4A-B). We examined this by comparing neural activity proceeding a state transition to activity earlier in a bout of a given behavior. On average, transitions from quiescence to running were preceded by a slight increase in residual neural activity (Fig. S4). However, we did not find evidence for a subpopulation of neurons that reliably encode state transitions; any neuron is equally likely to exhibit a large response during the transition from one behavior bout to another (Fig. S4).”

And the addition to Figure S4 and corresponding legend are:

“(A-C) Comparison of the change in neural activity at transitions from running to quiescence (left) and from quiescence to running (right). The “transition” period is the final half second before the transition, and the “preceding” period is the half second immediately before that. Change in activity per second for a given neuron is the average slope of activity over the time period, and the plotted quantity is the average for each neuron over all periods of a given type. (A) The average change in activity over all neurons is negative during transitions to quiescence (left) and positive during transitions to running (right); both shifts are small but significantly different than the values from the corresponding preceding periods ($p < .0001$, two-sample t-test). (B) Same as ‘A’ but shuffling cell identity before computing the slope of activity of each cell. For transitions to quiescence (left) and transitions to running (right), the effects seen in ‘A’ are maintained. (C) Comparison of transition periods in ‘A’ and ‘B’. Means are not significantly different (two-sample t-test).”

Minor points:

The behavior analysis is complex and employs many different tools. Please consider adding a flow diagram to help the reader.

We have added the following flow diagram to Figure S1:

It would be helpful to the reader to see how the recorded area relate to the whole brain when introducing the data. For example the authors could show the data on a template brain (like S2 F,G) earlier (in Fig 1 or S1).

We have added this reference cartoon to the corner of Figure 1G, with the corresponding update to the Figure 1 legend:

“(G) Sample volume of raw imaging data in a brain with pan-neuronal expression of both nuclear-localized GCaMP6s and nuclear dsRed. Shown are maximum-intensity projections of the dsRed channel over the approximate dorsal/ventral (top right), anterior/posterior (bottom right), and medial/lateral dimensions (top left). Pseudocolor indicates depth in dorsal/ventral dimension. Scale bar in spatial map is 50 μm . Cartoon at bottom left shows the approximate location of the imaged volume on a reference brain from a dorsal (top) and anterior (bottom) perspective.”

Line 56: “This provides a comprehensive picture of activity in the fly brain. “ This is misleading as the number of neurons is more than one order of magnitude smaller than the number of neurons in the brain.

We appreciate the reviewer’s catching this potentially misleading choice of words. We have changed this sentence to read:

“This provides an extensive picture of activity of individual neurons across the fly brain.”

Line 71: Please clarify here or in the methods whether voxel size (defined by the optics) is the same as spatial sampling rate.

We have added a reference to Methods in this sentence from the Results. The corresponding update to the Methods now reads:

“In raw data, the voxel size along two dimensions is isotropic and defined by the camera chip, while voxel size in the third dimension is the step size of scanning. Here, the scan dimension was anterior to posterior. Because the light sheet accesses the brain from an oblique angle, we orthogonalize the coordinate system before further processing, resulting in a typical voxel size of 1 x 1.4 x 2.4 μ m”

Fig2: The organization of this figure is confusing. Please consider re-organizing it to have all time constant together.

We unnecessarily drew attention to the time constants of the cells in Fig. 2A, when this panel was meant simply to show example fits more generally. The only figure panels related to time constants are already together (Fig. 4G-I). To alleviate this point of confusion, we have changed the caption of Fig. 4A to:

“Example traces from two cells, with regression fit overlaid in blue and ethogram below.”

Line 120: “Notably, this biologically meaningful heterogeneity in adjacent neurons would be masked by imaging the neuropil formed by these neurons.” I am not sure whether this is accurate as adjacent cell bodies could correspond to projections in different neuropil areas.

This text was meant to refer specifically to PI neurons, many of whom project to the same neuropil, but we did not make this sufficiently clear. We have replaced the above sentence with the following:

“Notably, many of these peptidergic PI neurons project to the same neuropil (De Velasco et al., 2007), meaning that this biologically meaningful heterogeneity in adjacent neurons would likely be masked in neuropil imaging.”

Line 253: The example provided don't reflect the scale down to a couple of neurons. Consider adding an example with just one or two neurons per neuron type.

We appreciate this suggestion. We have changed this sentence to the following: “This is consistent with known classes of cells in the fly brain - for example, Kenyon cells can be subdivided into alpha/beta, alpha'/beta', and gamma subclasses; dopaminergic neurons not only contain subclasses such as the PPL1 cluster but also sub-subclasses such as single PPL1 neurons that innervate many mushroom body compartments (Aso et al., 2014).”

Line 281: “Thus, most or perhaps all neurons that exhibit a negative correlation in panneuronal data are the peptidergic cell type Dilp.” Maybe add “in the dorsal brain” to clarify that there could be negatively correlated neurons elsewhere.

This sentence now reads:

“Thus, most or perhaps all neurons that exhibit a negative correlation in the dorsal brain are the peptidergic cell type Dilp.”

Fig5A: $\int_0^{2\pi} \left[\left(\sin \theta \right)^2 d\theta \right] = \int_0^{2\pi} \left[\left(\sin n\theta \right)^2 d\theta \right]$, so changing the frequency is maybe not the best way to convey the change that would be detected by the variance analysis.

Please consider changing the amplitude of the oscillations to better illustrate what is detected in the rest of the figure.

We have made this change to the middle panel of Figure 5A. Figure 5A now looks as follows:

Dilp and Dh44: A quantification of the comparison between the clusters found from large scale recording and the Dilp and Dh44 would help make the point that they are the same. For example is it possible to quantify the likelihood of these neuron types to be one of the clusters detected from the functional analysis, based on their position and number of neurons?

We would love to do this, but quantifying these likelihoods accurately would require an analysis of the statistics of cell type sizes as a function of spatial location across the brain, which is beyond the scope of this work. In the same spirit, we have undertaken an analysis to ask whether the statistics of running correlations among cells of a specific type are sufficient to identify where those cells reside in the brain. We have added the following to the final section of the Results, Figure S7, and the Figure S7 legend, respectively:

“Indeed, analysis of the distribution of running correlations observed in different parts of the brain confirmed that Dilp and Dh44 exhibit running correlations that one would only expect to find in PI (Fig. S7). Thus, these cell types are likely to correspond to unique clusters of neurons we identified in PI with panneuronal imaging.”

“(A) Wasserstein distance between the distribution of running correlations among Dilp neurons versus local patches of neurons imaged with the panneuronal driver. The most similar region (patch with the smallest distance) is indicated with an asterisk and falls within the PI region. The contours of high cell body density are overlaid in white for reference. (B) Distribution of running correlations among Dilp neurons versus the optimal local patch of neurons indicated by an asterisk in 'A'. (C) Same as 'A', but for Dh44 neurons. (D) Same as 'B', but for Dh44 neurons.”

Line 366: Please give more details on the dissection process. Are there exclusion criteria for flies that were damaged after surgery?

To address this and the subsequent point, we have added the following paragraph to the 'Mount and preparation' subsection of Methods:

“Our mounting and dissection procedure was very similar to prior work (Seelig et al., 2010) but with a larger dissected window to accommodate SCAPE (Fig. 1B); all dissections that opened up a window similar to Fig. 1B without damaging the brain were deemed successful. After dissection, flies were tested for robust behavior on the spherical treadmill - we defined robust behavior as exhibiting bouts of walking totalling at least one minute in a five minute span. All flies that exhibited robust behavior post-dissection were imaged. Most flies that passed these criteria continued to exhibit robust behavior for many minutes, but we only analyzed data from flies that exhibited bouts of walking totalling at least one minute in the first five minutes of imaging. Imaging continued for up to 30 minutes, terminating when a fly no longer exhibited bouts of walking. The mean experiment duration over all flies included in analysis was 18.1 minutes.”

Line 369: Please provide more information on the length of experiments.

Done. See previous comment.

Fig. 6, S6 and line 191, 247, 490: “folding the brain along the midline” this is confusing as it would result in flipping the z axis for one half of the brain.

We have changed this to “reflecting the brain along the midline.”

REVIEWERS' COMMENTS

Reviewer #1 (Remarks to the Author):

I thank the authors for their thorough responses to my comments and the additional clarifications and analyses performed. I believe this manuscript is much improved. While I wish to be pedantic about the v^2 vs. v point in the sparsity calculation, I do understand the authors rational and therefore will simply agree to have a different opinion. Therefore I am happy to recommend this manuscript for publication.

Reviewer #2 (Remarks to the Author):

We thank the authors for providing extra references. The references do suggest that the nuclear calcium follows neural activity, at least roughly. However, it is hard to agree that it is "faithful readouts of neural activity". It is clear from Weislogel et al., 2013 that even in the footshock case the dynamics seen in nuclear reporters were very different from the cytosolic reporters. And for the odor case, the nuclear reporters do not show any change. It might be that this is due to an older version of calcium indicators that are not sensitive, as the authors speculate, or it might be due to the speed of the calcium trafficking pathways into the nucleus. Thus, hedging the claim with "losing fast dynamics" is insufficient. The disadvantages of using nuclear reporters, that is changes in time courses, loss of small changes, need to be made clear. The phrase "faithful readouts of neural activity" makes it seem like there are very few disadvantages to it.

Another point that needs to be added to the discussion is the role of nuclear calcium in gene regulation, as the authors mentioned in their comment, and that the nuclear calcium readouts seen might be linked to the timescales of gene regulation rather than the timescales of neural activity.

Reviewer #3 (Remarks to the Author):

The responses provided by the authors have addressed my queries and concerns. I believe this paper will significantly contribute to the body of scientific knowledge.